# Glycogen deficiency enhances carbon partitioning into glutamate for an alternative extracellular metabolic sink in cyanobacteria

Yuichi Kato [1,2], Ryota Hidese[1,2], Mami Matsuda[1], Ryudo Ohbayashi [3], Hiroki Ashida[4], Akihiko Kondo [1,2,5] & Tomohisa Hasunuma [1,2] ✉

Glycogen serves as a metabolic sink in cyanobacteria. Glycogen deficiency causes the extracellular release of distinctive metabolites such as pyruvate and 2-oxoglutarate upon nitrogen depletion; however, the mechanism has not been fully elucidated. This study aimed to elucidate the mechanism of carbon partitioning in glycogen-deficient cyanobacteria. Extracellular and intracellular metabolites in a glycogen-deficient Δ*glgC* mutant of *Synechococcus elongatus* PCC 7942 were comprehensively analyzed. In the presence of a nitrogen source, the Δ*glgC* mutant released extracellular glutamate rather than pyruvate and 2-oxoglutarate, whereas its intracellular glutamate level was lower than that in the wild-type strain. The de novo synthesis of glutamate increased in the Δ*glgC* mutant, suggesting that glycogen deficiency enhanced carbon partitioning into glutamate and extracellular excretion through an unidentified transport system. This study proposes a model in which glutamate serves as the prime extracellular metabolic sink alternative to glycogen when nitrogen is available.

Glycogen, a highly branched polysaccharide, serves as a universal and major carbon storage compound in cyanobacteria that plays an important role in energy storage and is an endogenous metabolic substrate during nutrient starvation[1]. In cyanobacteria, glycogen synthesis and accumulation are induced by unfavorable environments, including depletion of nitrogen, phosphorus, and iron[2] and oxidative stress[3]. Starting from glucose 1-phosphate (G1P), cyanobacteria synthesize glycogen using enzymes including ADP-glucose pyrophosphorylase (also referred to as G1P adenylyl transferase) (GlgC), glycogen synthases (GlgA1/GlgA2), and 1,4-α-glucan branching enzyme (GlgB)[1,4]. In several cyanobacteria, such as *Synechocystis* sp. PCC 6803 (PCC 6803), *Synechococcus elongatus* PCC 7942 (PCC 7942), and *Synechococcus* sp. PCC 7002 (PCC 7002), glycogen synthesis and accumulation were completely abolished by either single inactivation of the *glgC* gene or simultaneous inactivation of the *glgA1*/*glgA2* genes[5–7]. Although glycogen-deficient mutants are sensitive to several adverse conditions, such as high light, light-dark regimes, low

inocula, and nitrogen starvation, and show reduced photosynthetic capacity and a non-bleaching phenotype upon nitrogen depletion[8–10], they can still grow photoautotrophically by fixing atmospheric carbon dioxide ($CO_2$).

Cyanobacteria can directly convert $CO_2$ into valuable compounds using sunlight as an energy source and have been studied as sustainable producers of various biochemicals[11]. Blocking glycogen synthesis has been considered as a promising general approach for improving carbon partitioning into the desired compounds. For example, glycogen deficiency caused by the inactivation of both *glgA1* and *glgA2* genes increased the yield of mannitol in PCC 7002[12]. In PCC 6803, ethanol production was improved by the deletion of *glgC*, particularly when the cells were cultured without nitrogen[13]. Strengthening glycogen degradation activity also improved the production titer; overexpression of glycogen phosphorylase GlgP, which degrades glycogen into G1P, decreased glycogen accumulation and increased the secretory production of sucrose in PCC 7942[14]. These reports

---

[1]Engineering Biology Research Center, Kobe University, 1-1 Rokkodai, Nada, Kobe 657-8501, Japan. [2]Graduate School of Science, Technology and Innovation, Kobe University, 1-1 Rokkodai, Nada, Kobe 657-8501, Japan. [3]Department of Biological Science, Faculty of Sciences, Shizuoka University, 836 Ohya, Suruga, Shizuoka 422-8529, Japan. [4]Graduate School of Human Development and Environment, Kobe University, 3-11 Tsurukabuto, Nada, Kobe 657-8501, Japan. [5]Department of Chemical Science and Engineering, Graduate School of Engineering, Kobe University, 1-1 Rokkodai, Nada, Kobe 657-8501, Japan. ✉e-mail: hasunuma@port.kobe-u.ac.jp

indicate that modification of glycogen synthesis and degradation has a large impact on carbon metabolism in cyanobacteria.

The extracellular release of distinctive metabolites in glycolysis and the tricarboxylic acid (TCA) cycle upon nitrogen depletion is a noteworthy feature of glycogen-deficient cyanobacteria. This metabolic phenomenon occurs because they partition fixed carbon into glycolysis and the TCA cycle, in addition to converting it into alternative storage compounds, such as glucosylglycerol and sucrose, in the *glgA1/A2* mutant[15]. For example, the Δ*glgC* mutant of PCC 7942 exhibited higher intracellular levels of 3-phosphoglycerate (3-PGA), citrate (Cit), succinate (Suc), and 2-oxoglutarate (2-OG) than the wild type and excreted pyruvate (Pyr), Suc, fumarate (Fum), and 2-OG[9,16]. In glycogen-deficient mutants of PCC 6803, Pyr and 2-OG overflow extracellularly in a light-dependent manner[8,17]. In PCC 7002, the extracellular release of Pyr, Suc, acetate, 2-OG, and α-ketoisocaproate is increased in the Δ*glgC* mutant[10,18]. Thus, the extracellular release of glycolytic and TCA cycle metabolites, especially Pyr and 2-OG, is a common phenotype of glycogen-deficient cyanobacteria upon nitrogen depletion and light illumination. In addition, the glycogen-deficient mutant exhibited a high energy charge under high-light conditions, suggesting that glycogen synthesis functions as an energy buffer in cyanobacteria[19]. Therefore, the extracellular release of metabolites has been suggested as an alternative dissipation mechanism for excess energy in glycogen-deficient cyanobacteria when photosynthetic carbon fixation exceeds biomass accumulation. However, the reasons for the release of glycolytic and TCA cycle metabolites have not been fully elucidated. In addition, how nitrogen depletion is involved in the metabolite release remains unclear.

The present study aimed to elucidate the carbon-partitioning mechanism of glycogen-deficient cyanobacteria by investigating the extracellular and intracellular metabolic consequences. To examine why the release of Pyr and 2-OG occurs only under nitrogen-depleted conditions, metabolomic analysis was performed on the Δ*glgC* mutant of PCC 7942 cultured in the presence of a nitrogen source. High levels of glutamate (Glu), rather than Pyr or 2-OG, were found to be extracellularly released in the presence of a nitrogen source. In addition, the de novo synthesis of Glu increased in the Δ*glgC* mutant in the presence of a nitrogen source. Thus, it was suggested that, when a nitrogen source is available, glycogen deficiency enhances carbon partitioning into Glu, which serves as the prime extracellular metabolic sink alternative to glycogen.

## Results

### Metabolites extracellularly released from the glycogen-deficient cells

To investigate the metabolic consequences of glycogen-deficiency in the absence and presence of a nitrogen source, a Δ*glgC* mutant strain of PCC 7942 was cultured using the BG-11 medium with initial nitrate concentrations of 7.5 mM (low nitrogen, LN) and 17.6 mM (high nitrogen, HN; the standard nitrate concentration of BG-11 medium). The nitrate concentration in the culture medium was analyzed, and under LN conditions, PCC 7942 (wild type) and the Δ*glgC* mutant completely consumed nitrate on days 2 and 5, respectively (Fig. 1a), showing that these strains were in nitrogen depletion after day 5. Under HN conditions, in contrast, nitrate in the medium remained available at day 10, indicating that these strains were in nitrogen repletion during cultivation. The phosphate concentrations in the culture medium were also analyzed, and it was discovered that these strains completely consumed phosphate by day 3 under both LN and HN conditions (Fig. 1b). The Δ*glgC* mutant exhibited significantly lower biomass accumulation than did the wild type under LN and HN conditions (Fig. 1c). Under LN conditions, the biomass accumulation of the Δ*glgC* mutant at day 10 was 0.86 g-dry cell weight (DCW)·L⁻¹, which was 43.2% of that of the wild type (1.98 g-DCW·L⁻¹). Similarly, under HN conditions, the biomass accumulation of the Δ*glgC* mutant at day 10 was 1.59 g-DCW·L⁻¹, which was 52.9% of that of the wild type (3.01 g-DCW·L⁻¹). The glycogen content in the wild-type biomass started to increase from day 3 under both nitrogen conditions (Fig. 1d). During the 10-day cultivation, the glycogen content of the wild-type cells reached 47.4% of DCW under LN conditions and 35.2% of DCW under HN conditions. In contrast, no glycogen was detected in the Δ*glgC* mutant cells under either nitrogen condition. The significantly lower biomass accumulation in the Δ*glgC* mutant than in the wild type would be caused by deficiency of glycogen, which is the major biomass component in PCC 7942, as well as the extracellular release of metabolites, as previously reported[9,16].

To investigate the extracellularly released metabolites under these nitrogen conditions, the metabolites in the culture supernatant were

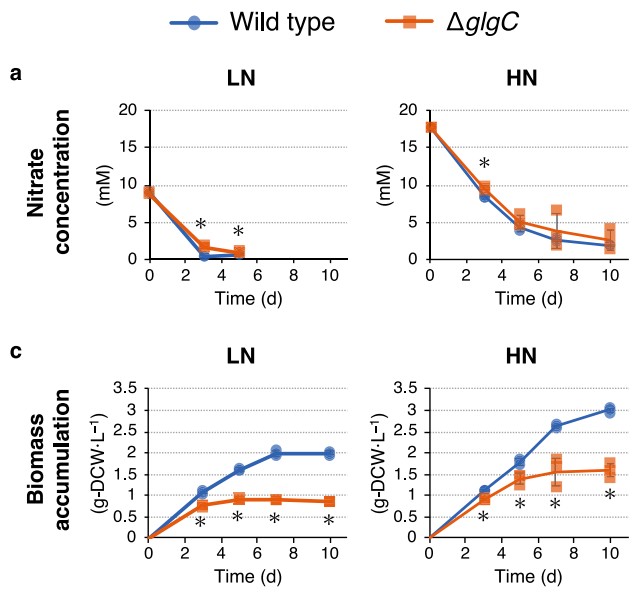

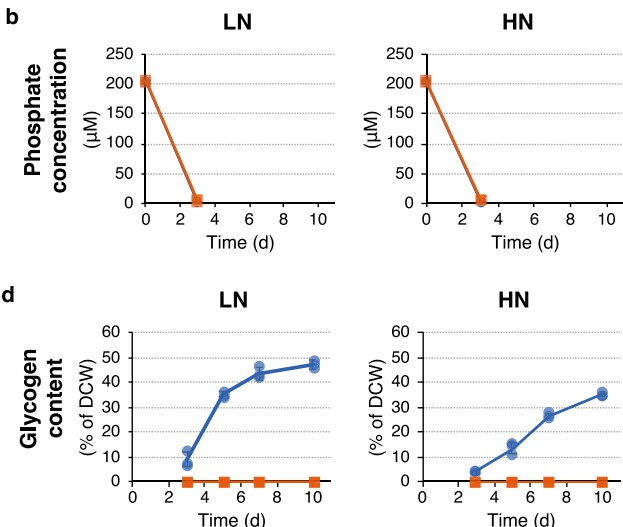

**Fig. 1 | Evaluation of the Δ*glgC* mutant cyanobacteria for nitrate consumption, phosphate consumption, biomass accumulation, and glycogen accumulation.** The wild-type (blue circles) and Δ*glgC* mutant (orange squares) cyanobacteria were cultured using media with an initial nitrate concentration of 7.5 mM (low nitrogen, LN) and 17.6 mM (high nitrogen, HN). Subsequently, nitrate concentration in medium (**a**), phosphate concentration in medium (**b**), dry cell weight (DCW)-based biomass accumulation (**c**), and cellular glycogen content (**d**) were analyzed. Values are shown as the mean ± standard deviation of three replicate experiments (*$P < 0.05$ by Welch's *t* test).

comprehensively analyzed. Under LN conditions, the $\Delta glgC$ mutant released high amounts (more than 10 mg·L$^{-1}$ at day 10) of the metabolites involved in glycolysis and the TCA cycle (i.e., Pyr, 2-OG, and Suc) into the medium (Fig. 2a). The release of 2-OG and Suc was consistent with a previous report on the $\Delta glgC$ mutant of PCC 7942[9]. Additionally, the release of Pyr and 2-OG has been reported on the $\Delta glgA1/glgA2$ mutant and $\Delta glgC$ mutant in PCC 6803[8,17,20], and release of 2-OG, Pyr, and Suc has been reported on the $\Delta glgC$ mutant in PCC 7002[18], indicating that this phenomenon is not specific to PCC 7942. In the present study, the $\Delta glgC$ mutant cells particularly released Pyr (41.91 mg·L$^{-1}$ at day 10) and 2-OG (506.27 mg·L$^{-1}$ at day 10), which were more than 100-fold higher than those of the wild type. The $\Delta glgC$ mutant also released more lactate (Lac) than the wild type under both nitrogen conditions. Under HN conditions, in contrast, the $\Delta glgC$ mutant released 3.76 mg·L$^{-1}$ Pyr and 9.88 mg·L$^{-1}$ 2-OG, which was significantly lower than that under LN conditions. Instead of these metabolites in glycolysis and the TCA cycle, the $\Delta glgC$ mutant released high amounts of several proteinogenic amino acids (i.e., alanine and glutamate) under HN conditions (Fig. 2b). In particular, the $\Delta glgC$ mutant released Glu at 81.44 mg·L$^{-1}$ after 10 days of cultivation. Thus, in the presence of a nitrogen source, glycogen deficiency caused the extracellular release of Glu rather than Pyr or 2-OG.

### Levels of intracellular metabolites in the glycogen-deficient cells

To elucidate the mechanism of metabolite release in the $\Delta glgC$ mutant, the intracellular metabolite levels were comprehensively analyzed. Regarding the Calvin–Benson–Bassham (CBB) cycle, the $\Delta glgC$ mutant showed an increased accumulation of 3-PGA, whereas the levels of sedoheptulose-7-phosphate (S7P) and ribulose-5-phosphate (Ru5P) decreased under LN conditions after nitrogen depletion on day 2 (Fig. 3a). This suggested that the conversion of 3-PGA into the intermediates in the CBB cycle decreased in the $\Delta glgC$ mutant. In addition, the $\Delta glgC$ mutant showed a decrease in adenosine triphosphate (ATP) levels on days 5 and 7 under LN conditions together with significantly lower levels of adenosine diphosphate (ADP) compared to the wild type under both nitrogen conditions. The levels of 2-phosphoglycerate (2-PGA) and phosphoenolpyruvate (PEP) in the $\Delta glgC$ mutant were lower than those in the wild type under both LN and HN conditions, suggesting that carbon partitioning from these metabolites was enhanced toward downstream metabolites in the mutant. The levels of Lac in the $\Delta glgC$ mutant cells were also lower than in the wild type under both LN and HN conditions. Because the extracellular levels of Lac were higher in the $\Delta glgC$ mutant compared to those in the wild type (Fig. 2a), it is possible that the extracellular release of Lac was activated in the mutant. The extracellularly released metabolites other than Lac in the $\Delta glgC$ mutant, i.e., Pyr, 2-OG, Suc, and Mal, accumulated intracellularly under LN conditions (Fig. 3b). This result is mostly consistent with that of a previous study reporting that intracellular levels of 3-PGA, Cit, Suc, and 2-OG increased in the $\Delta glgC$ mutant of PCC 7942 upon nitrogen starvation[9]. Since the intracellular levels of 2-OG, Suc, Mal, and Pyr were higher and the level of acetyl-coenzyme A (AcCoA) was lower than that in the wild type after day 5 under LN conditions, conversion of Pyr into AcCoA might be the rate-limiting step in the $\Delta glgC$ mutant (Fig. 3b). Under HN conditions, the intracellular accumulation of Pyr, 2-OG, Suc, and Mal was not significantly different between the wild type and $\Delta glgC$ mutant, compared to that under LN conditions. The intracellular level of Glu significantly decreased in the $\Delta glgC$ mutant under both nitrogen conditions. As the extracellular level of Glu was significantly higher in the $\Delta glgC$ mutant than in the wild type (Fig. 2b), an export mechanism for Glu was suggested to be activated in the mutant.

### Analysis of newly synthesized metabolites from CO$_2$

As the $\Delta glgC$ mutant extracellularly released Pyr, 2-OG, and Glu more than the wild type (Fig. 2a, b), it was hypothesized that the de novo synthesis ratio of these metabolites would increase in the mutant. To test this hypothesis, metabolites newly synthesized by carbon fixation were analyzed using dynamic metabolic analysis. Newly synthesized metabolites were labeled with $^{13}$C by adding NaH$^{13}$CO$_3$ to the culture medium on day 5 when the

nitrate in the medium was completely depleted under LN conditions (Fig. 1a). Subsequently, the time course changes in the labeling ratio of the intracellular metabolites were analyzed. However, Pyr data were not obtained because of the low signal abundance. Compared to that in the wild type, the labeling ratio of 3-PGA in the $\Delta glgC$ mutant remained unchanged and decreased under LN and HN conditions, respectively (Fig. 4). This suggests that glycogen deficiency causes a decrease in carbon fixation under nitrogen-replete conditions but not under nitrogen-depleted conditions. During glycolysis, the labeling ratio of 2-PGA under LN conditions increased in the $\Delta glgC$ mutant. Under HN conditions, the labeling ratio of 2-PGA did not change in the $\Delta glgC$ mutant, whereas that of the upstream metabolite 3-PGA decreased. In addition, the labeling ratio of PEP increased in the $\Delta glgC$ mutant under both LN and HN conditions. These results suggest that carbon partitioning into glycolysis is enhanced by glycogen deficiency independent of nitrogen availability. In the $\Delta glgC$ mutant, the labeling ratio of 2-OG significantly increased under both LN and HN conditions. Furthermore, the labeling ratios of Glu and glutamine (Gln) increased independent of nitrogen conditions. Compared with those under HN conditions, $^{13}$C fractions of Glu and Gln in the $\Delta glgC$ mutant rapidly increased under LN conditions, possibly because of low intracellular levels. The intracellular levels of 2-OG, Glu, and Gln during the $^{13}$C-labeling experiment were also analyzed (Supplementary Fig. 1a), and found to be consistent with the result shown in Fig. 3b. Based on these data, the levels of newly synthesized and intracellularly accumulated metabolites were calculated. The intracellular levels of $^{13}$C-labeled 2-OG and $^{13}$C-labeled Glu were significantly higher in the $\Delta glgC$ mutant than they were in the wild type under LN and HN conditions (Supplementary Fig. 1b). These results suggest that glycogen deficiency enhances the partitioning of newly fixed carbon into the TCA cycle, or further into the Gln synthase (GS)-Glu synthase (GOGAT) cycle when a nitrogen source is available.

### Discussion

Glycogen synthesis functions as a metabolic sink in cyanobacteria when the amount of photosynthetically fixed carbon exceeds the amount used for cellular maintenance and growth. In glycogen-deficient mutant cyanobacteria, metabolites such as Pyr and 2-OG are released extracellularly upon nitrogen depletion and serve as alternative sinks for excess light energy[8–10,16–18]. The present study investigated the metabolic consequence of glycogen deficiency under nitrogen-depleted and -repleted conditions by culturing the glycogen-deficient $\Delta glgC$ mutant in PCC 7942 under LN and HN conditions, respectively. Under LN conditions, the wild type and $\Delta glgC$ mutant were considered nitrogen-depleted after day 5 because they had completely consumed the nitrate in the medium by that timepoint. Under HN conditions, these strains were considered to be in nitrogen repletion throughout the cultivation since the nitrate in the medium remained available even on day 10. In addition, under HN conditions, the wild-type cells accumulated glycogen, even though the nitrogen source was not depleted (Fig. 1a, d). Since phosphate in the culture medium was completely depleted by day 3 (Fig. 1b), the glycogen accumulation under HN conditions was suggested to be induced by phosphorus depletion[2]. The glycogen accumulation in wild-type cells might be also enhanced by high CO$_2$ (1%) conditions since cyanobacterial glycogen content was reported to be increased under 1%, 2%, and 4% CO$_2$ compared to that under 0.04% CO$_2$ conditions[21]. The present study showed that the glycogen-deficient $\Delta glgC$ mutant in PCC 7942 releases Glu extracellularly rather than Pyr and 2-OG when nitrogen is available (Fig. 2). Since Glu is an abundant metabolite in cells (Fig. 3), cell lysis during cultivation or cell disruption when preparing extracellular supernatants via centrifugation might occur in the $\Delta glgC$ mutant. Intracellular metabolites were detected in the $\Delta glgC$ mutant, and several metabolites were detected at higher levels than in the wild type (Fig. 3), suggesting that the cells were not lysed, at least completely, during cultivation. In addition, the extracellularly detected Glu in the $\Delta glgC$ mutant was highly abundant compared to that detected intracellularly. Under the HN conditions on day 10, for example, 81.44 mg·L$^{-1}$ Glu was extracellularly detected in the $\Delta glgC$ mutant, while the amount of Glu in the

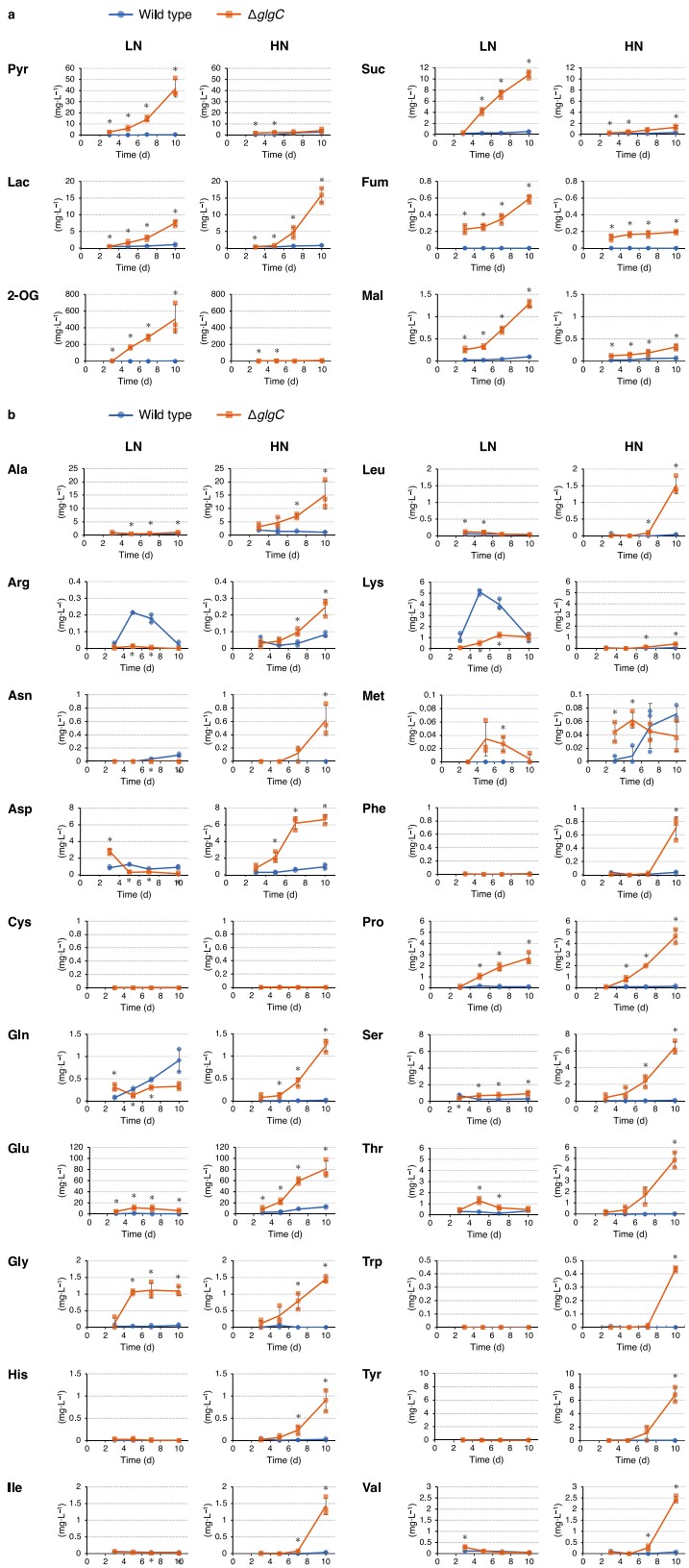

**Fig. 2 | Analysis of extracellularly released metabolites from the ΔglgC mutant cyanobacteria.** The wild-type (blue circles) and ΔglgC mutant (orange squares) cyanobacteria were cultured using media with the initial nitrate concentration of 7.5 mM (low nitrogen, LN) and 17.6 mM (high nitrogen, HN). Subsequently, extracellularly released metabolites in glycolysis and the TCA cycle (**a**) and proteinogenic amino acids (**b**) were analyzed. Pyr pyruvate, Lac lactate, 2-OG

2-oxoglutarate, Suc succinate, Fum fumarate, Mal malate, Ala alanine, Arg arginine, Asn asparagine, Asp aspartate, Cys cysteine, Gln glutamine, Glu glutamate, Gly glycine, His histidine, Ile isoleucine, Leu leucine, Lys lysine, Met methionine, Phe phenylalanine, Pro proline, Ser serine, Thr threonine, Trp tryptophan, Tyr tyrosine, Val valine. Values are shown as the mean ± standard deviation of three replicate experiments (*$P < 0.05$ by Welch's $t$ test).

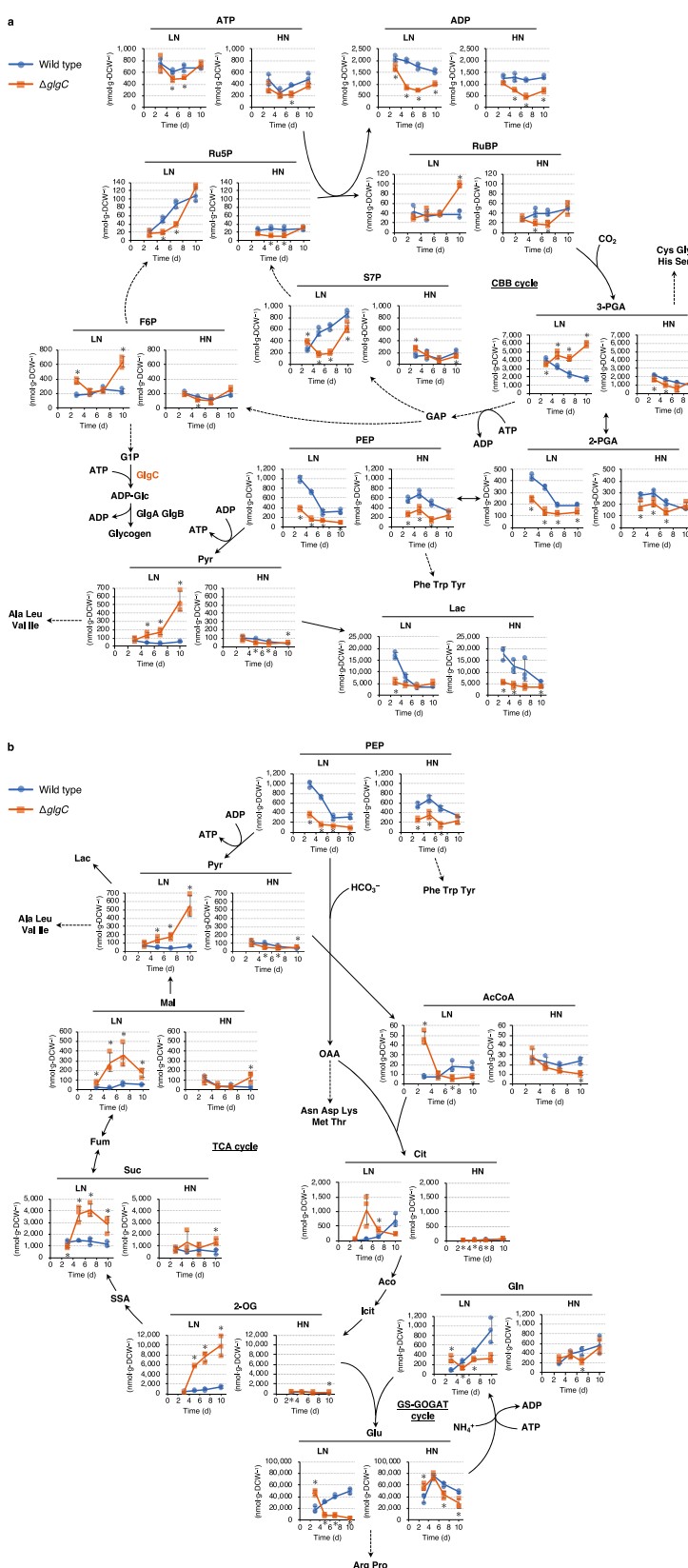

cells was 6.64 mg·L⁻¹, which was calculated from the intracellular level of Glu (28,365.21 nmol·g-DCW⁻¹) and the accumulated biomass (1.59 g-DCW·L⁻¹, Fig. 1c). This indicates that most of the extracellularly detected Glu in the ΔglgC mutant would be continuously released from viable cells during cultivation rather than being derived from cell disruption or residual cells when preparing extracellular metabolites. As it was also

observed in wild-type cells, the extracellular release of Glu might be an inherent ability of carbon partitioning in response to environmental fluctuations. In addition to glycogen, several cyanobacteria accumulate poly-hydroxybutyrate (PHB) upon nitrogen depletion. The *sll0783* (nitrogen starvation response protein) mutant of PCC 6803, in which PHB accumulation decreases, showed a higher level of fructose-6-phosphate (F6P)

**Fig. 3 | Analysis of intracellular metabolites in the Δ*glgC* mutant cyanobacteria.** The wild-type (blue circles) and Δ*glgC* mutant (orange squares) cyanobacteria were cultured using media with initial nitrate concentrations of 7.5 mM (low nitrogen, LN) and 17.6 mM (high nitrogen, HN); then, the intracellular metabolites were comprehensively analyzed. The solid and dotted lines represent single and multiple enzymatic steps, respectively. **a** Metabolites in the CBB cycle and glycolysis, **b** metabolites in the glycolysis, TCA cycle, and GS-GOGAT cycle. 2-OG 2-oxoglutarate, 2-PGA 2-phosphoglycerate, 3-PGA 3-phosphoglycerate, AcCoA acetyl-coenzyme A, Aco aconitate, ADP adenosine diphosphate, ADP-Glc adenosine diphosphate-glucose, Ala alanine, Arg arginine, Asn asparagine, Asp aspartate, ATP adenosine triphosphate, CBB cycle Calvin–Benson–Bassham cycle, Cit citrate, Cys cysteine, DCW dry cell weight, F6P fructose-6-phosphate, Fum fumarate, G1P glucose-1-phosphate, GAP glyceraldehyde-3-phosphate, Gln glutamine, GS-GOGAT cycle glutamine synthase–glutamate synthase cycle, Glu glutamate, Gly glycine, His histidine, Icit isocitrate, Ile isoleucine, Lac lactate, Leu leucine, Lys lysine, Mal malate, Met methionine, OAA oxaloacetate, PEP phosphoenolpyruvate, Phe phenylalanine, Pro proline, Pyr pyruvate, Ru5P ribulose-5-phosphate, RuBP ribulose-1,5-bisphosphate, S7P sedoheptulose-7-phosphate, Ser serine, SSA succinic semialdehyde, Suc succinate, TCA cycle tricarboxylic acid cycle, Thr threonine, Trp tryptophan, Tyr tyrosine, Val valine. Error bars indicate standard deviation of three replicate experiments (*$P < 0.05$ by Welch's *t* test).

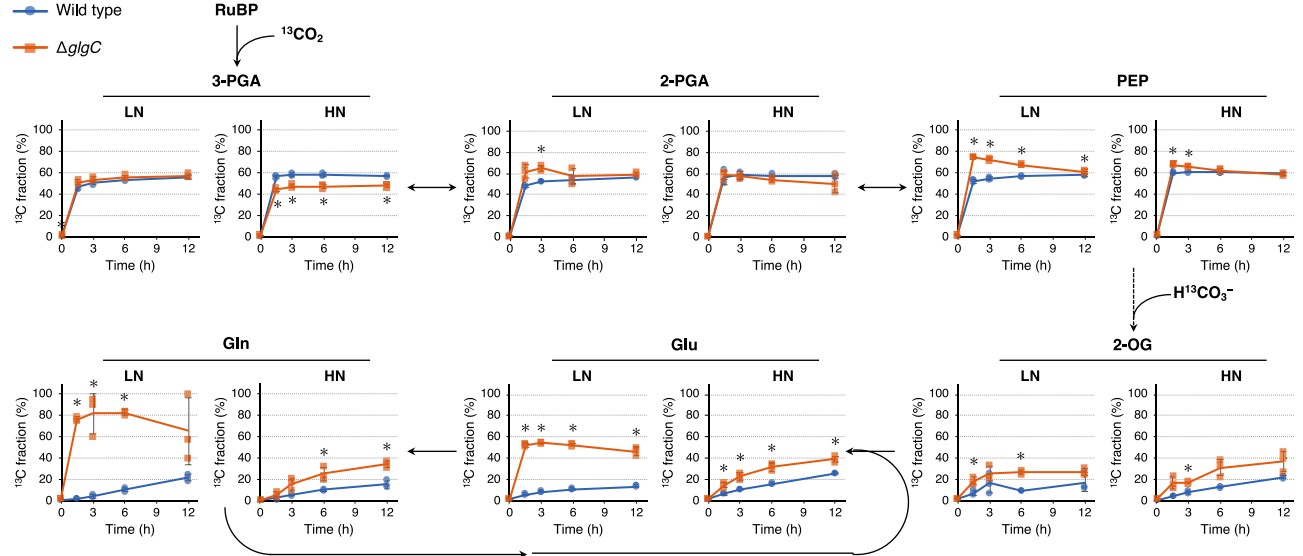

**Fig. 4 | Time course of $^{13}C$ labeling of the newly synthesized metabolites.** The wild-type (blue circles) and Δ*glgC* mutant (orange squares) cyanobacteria were cultured using media with initial nitrate concentrations of 7.5 mM (low nitrogen, LN) and 17.6 mM (high nitrogen, HN) for 5 days. Subsequently, 25 mM NaH$^{13}CO_3$ was added to the culture medium as a carbon source. After 0 to 12 h cultivation, intracellular metabolites were analyzed, and the ratio of $^{13}C$ in the total carbon ($^{13}C$ fraction) was determined. The solid and dotted lines represent single and multiple enzymatic steps, respectively. 2-OG 2-oxoglutarate, 2-PGA 2-phosphoglycerate, 3-PGA 3-phosphoglycerate, Glu glutamate, Gln glutamine, PEP phosphoenolpyruvate, RuBP ribulose-1,5-bisphosphate. Values are shown as the mean ± standard deviation of three replicate experiments (*$P < 0.05$ by Welch's *t* test).

before nitrogen depletion and lower levels of PEP compared to the wild type[22], which is consistent with the present study (Fig. 3a). In contrast, changes in 3-PGA, AcCoA, Cit, 2-OG, Suc, and Glu observed in the *sll0783* mutant are not consistent with those in the Δ*glgC* mutant of PCC 7942 (Fig. 3a, b). One of the reasons might be that the *sll0783* mutant still accumulated glycogen and a small amount of PHB in the cells[23].

The de novo synthesis ratio of 2-OG, Glu, and Gln from CO$_2$ increased in the Δ*glgC* mutant under both LN and HN conditions (Fig. 4). The Δ*glgC* mutant showed high intracellular levels of 2-OG, a precursor of the GS-GOGAT cycle, under LN conditions (Fig. 3b), probably because the conversion of 2-OG into Glu was strongly restricted due to the unavailability of a nitrogen source (Fig. 5). In addition to 2-OG, accumulation of its downstream metabolites (i.e., Suc, Mal, and Pyr) in the Δ*glgC* mutant (Fig. 3b) suggested that conversion of Pyr into AcCoA was the rate-limiting step under LN conditions. As a result, Pyr and 2-OG would be excessively accumulated in the mutant, triggering the metabolic overflow into the medium under LN conditions (Fig. 5b). A previous study suggested the involvement of an unknown Pyr/proton symporter as high pH causes an increase in the extracellular release of Pyr[16]. In contrast, the intracellular level of Glu in the Δ*glgC* mutant was significantly lower than that in the wild type under LN conditions (Fig. 3b). Under HN conditions, the intracellular level of Glu in the Δ*glgC* mutant was not higher than that in the wild type after day 5, although it was released continuously during the culture period. These results suggest that Glu is released extracellularly via an export mechanism

activated by glycogen deficiency (Fig. 5a). A possible exporter of Glu is mechanosensitive channels (MSCs). In bacteria such as *Escherichia coli* and *Corynebacterium glutamicum*, MSCs are known to export osmolytes, including Glu, upon osmotic downshift[24,25]. Homology search revealed that PCC 7942 harbors three putative MSC genes; the large conductance mechanosensitive channel protein gene *mscL* (Synpcc7942_1991) homologous to *E. coli mscL* gene, the mechanosensitive ion channel family protein gene (Synpcc7942_0610) homologous to *E. coli ybiO* gene, and the mechanosensitive ion channel gene (Synpcc7942_0664) homologous to *E. coli mscM, mscS*, and *mscK* genes. Although the role of MSCs in cyanobacteria has not been thoroughly examined, the *mscL* gene in PCC 6803 has been found to contribute to hypoosmotic stress adaptation and Ca$^{2+}$ transport[26,27]. In the present study, the increased extracellular levels and decreased intracellular levels of Lac in the Δ*glgC* mutant also supported the hypothesis that MSCs are activated by glycogen deficiency (Figs. 2a and 3a).

The synthesis of Pyr and 2-OG is hypothesized to function as an alternative dissipation mechanism for excess light energy when glycogen synthesis is deficient. In a previous study, the energy charge (defined as the ratio of ATP to ADP + ATP) was higher in a glycogen-deficient mutant than in PCC 6803[19]. This is consistent with the results of the present study under both LN and HN conditions, largely because of the low ADP levels in the Δ*glgC* mutant (Fig. 3a). In addition, this study found that de novo synthesis and extracellular release of Glu increased in the Δ*glgC* mutant (Figs. 2b and 4, and Supplementary Fig. 1b), suggesting that the GS-GOGAT

**Fig. 5 | Alternative metabolic sink model of glycogen-deficient cyanobacteria.** A deficiency in glycogen synthesis enhances the partitioning of carbon into Glu and its excretion via an unidentified transport mechanism (**a**). When the nitrogen source is unavailable, Glu synthesis is strongly restricted, which results in intracellular accumulation and extracellular overflowing of 2-OG, the precursor of Glu, and Pyr (**b**). Thus, Glu, 2-OG, and Pyr serve as the extracellular metabolic sink alternative to glycogen when its synthesis is deficient. Bold arrows indicate processes enhanced by glycogen deficiency. Grey arrows indicate abolished or restricted processes. 2-OG 2-oxoglutarate, 3-PGA 3-phosphoglycerate, Glu glutamate, Gln glutamine, Mal malate, Suc succinate.

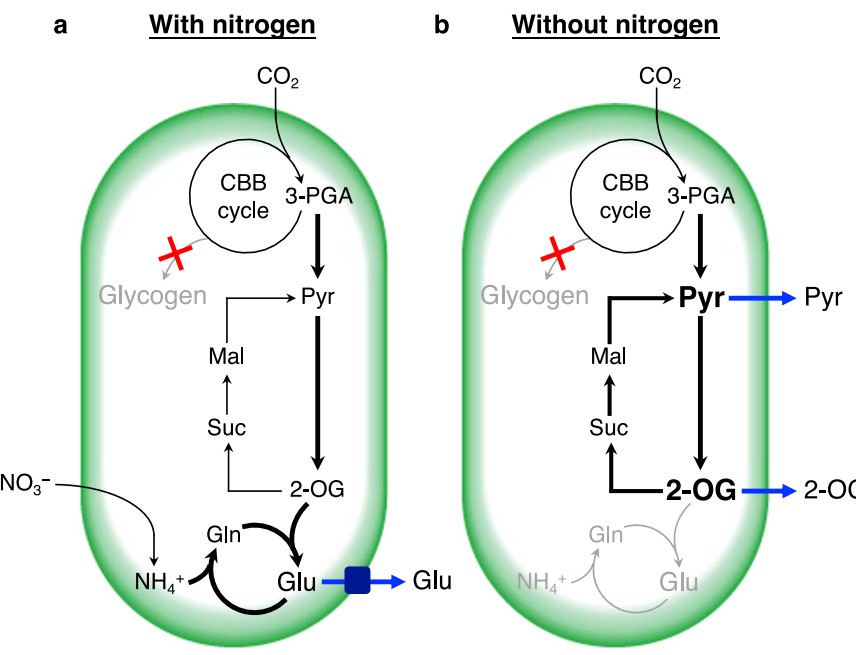

cycle, not Pyr or 2-OG, was the actual alternative metabolic sink during glycogen deficiency. The conversion of nitrate to Glu is an energetically expensive process that consumes 10 electrons[10,28]. Therefore, Glu is a more suitable product for energy dissipation than Pyr or 2-OG when a nitrogen source is available. In the present study, nitrate consumption of the Δ*glgC* mutant was similar to that of the wild type (Fig. 1a), suggesting that energy consumption via nitrogen assimilation was similar in the mutant. As another possible explanation, the present study proposes that Glu is released extracellularly for osmotic regulation, although further studies are required to gain a precise understanding of this metabolic phenomenon. Glycogen-deficient mutant cells are assumed to be under constitutive hypoosmotic stress because they accumulate fixed carbons as low-molecular-weight metabolites (Fig. 3) instead of as high-molecular-weight glycogen (Fig. 1d). Therefore, they may activate MSCs that cause extracellular release of Glu as an osmolyte[24], and consequently, carbon partitioning into the TCA cycle and GS-GOGAT cycle is enhanced for the supplementation of Glu. In conclusion, the present study suggests that glycogen deficiency in cyanobacteria enhances carbon partitioning into Glu, which serves as the prime extracellular metabolic sink alternative to glycogen.

## Methods
### Strains and culture conditions
The cyanobacterium *Synechococcus elongatus* PCC 7942 and its Δ*glgC* mutant strain were used in this study. For the analysis, the cyanobacteria were phototrophically cultured in double-deck flasks on a BR-40LF bioshaker (TAITEC, Aichi, Japan). The upper stage of the flasks was supplemented with 70 mL of BG-11 ($2.00 \times 10^{-2}$ M 4-(2-hydroxyethyl)-1-piperazineethanesulfonic acid (HEPES)-KOH (pH = 7.8), $3.04 \times 10^{-4}$ M MgSO$_4$·7H$_2$O, $2.58 \times 10^{-4}$ M CaCl$_2$·2H$_2$O, $2.24 \times 10^{-4}$ M K$_2$HPO$_4$, $1.89 \times 10^{-4}$ M Na$_2$CO$_3$, $4.63 \times 10^{-5}$ M H$_3$BO$_3$, $3.12 \times 10^{-5}$ M citric acid, $9.15 \times 10^{-6}$ M MnCl$_2$·4H$_2$O, $2.69 \times 10^{-6}$ M Na$_2$EDTA·2H$_2$O, $7.65 \times 10^{-7}$ M ZnSO$_4$·7H$_2$O, $3.20 \times 10^{-7}$ M CuSO$_4$·5H$_2$O, $1.70 \times 10^{-7}$ M Co(NO$_3$)$_2$·5H$_2$O, $8.68 \times 10^{-8}$ M Na$_2$MoO$_4$·2H$_2$O, and $6.00 \times 10^{-3}$ g·L$^{-1}$ Fe(III) ammonium citrate) containing 17.6 mM or 7.5 mM NaNO$_3$. The lower stage of the flasks was supplemented with 50 mL of 2 M K$_2$CO$_3$/KHCO$_3$ solution, which adjusted the CO$_2$ gas concentration to 1% (*v/v*). Cells were inoculated at an optical density of 750 nm (OD$_{750}$) = 0.1 and cultured under continuous illumination with white fluorescent lamps at 110–120 µmol photons·m$^{-2}$·s$^{-1}$ at 30 °C with rotary shaking at 100 rpm[29].

### Measurement of nitrate and phosphate
The culture was centrifuged at $8000 \times g$ for 5 min to prepare the clear supernatant without cells. To determine the nitrate concentration using a calibration curve, the optical density of the supernatant was measured at 220 nm (OD$_{220}$) using a UV mini-1240 UV–Vis spectrophotometer (Shimadzu, Kyoto, Japan)[30]. The phosphate concentration was determined using a PiBlue Phosphate Assay Kit (BioAssay Systems, Hayward, CA, USA) following the manufacturer's instructions.

### Construction of recombinant strain
The homology arms for the *glgC* gene (Synpcc7942_0603) were amplified from the PCC 7942 genomic DNA by PCR using the following primer pairs: upstream: 5′-AGTGAATTCGAGCTCGGTACCCCAGCGATCCGTGTCCCTACTC-3′ and 5′- CAATCTCCCCCAAGTCAAGCGG-3′; downstream: 5′-CACCATGCGCCTCGGCAAAG-3′ and 5′-GACCATGATTACGCCCTGCAGCAATTGCCCTAAGACAGTTGTCGTCTTTC-3′. Homology arms and a gentamycin acetyltransferase gene cassette were cloned into a plasmid vector using an In-Fusion HD Cloning Kit (Takara Bio USA, Inc., Mountain View, CA, USA). To construct a recombinant strain harboring Δ*glgC* mutation, PCC 7942 cells were cultured in BG-11 medium to the mid-exponential phase (OD$_{750}$ = 1.0, approximately), washed once with the medium, and resuspended with 10 times concentration in the medium. Next, 100 µL of the cell suspension was mixed with 1 µg of the plasmid. After slow rotation in the dark overnight, the mixture was spread onto a 0.45 µm pore size nitrocellulose filter (Merck Millipore, Burlington, MA, USA) placed on a BG-11 agar plate. The agar plates were incubated at 30 °C under continuous illumination using white fluorescent lamps. After 2 days, the filter attached with cells was transferred onto a fresh BG-11 agar plate containing 2 mg·L$^{-1}$ gentamicin for further cultivation and selection. Single colonies were individually cultured in BG-11 medium, and complete segregation of the Δ*glgC* mutant was confirmed using PCR of genomic DNA using the primer pair 5′-CCAGCGATCCGTGTCCCTACTCG-3′ and 5′-CAATTGCCCTAAGACAGTTGTCGTCTTTC-3′[31].

### Measurement of glycogen
Cells were harvested via centrifugation at $8000 \times g$ for 5 min, washed once with 20 mM ammonium carbonate, and lyophilized. Dried cells (5 mg) were suspended in 100 µL of 30% (*w/v*) KOH and incubated at 90 °C for 90 min. Next, 300 µL of ethanol (pre-cooled at 4 °C) was added, and the sample was

**Article**

mixed by vortexing and incubated on ice for 1 h. After centrifugation at $7500 \times g$ at 4 °C for 5 min, the resulting pellet was washed twice with 300 μL of ethanol (pre-cooled at 4 °C) and dried at 80 °C for 30 min. Glycogen was extracted by adding 100 μL of water and incubating under 1800 rpm agitation at 25 °C for 10 min. After centrifugation at $14,000 \times g$ for 5 min, glycogen in the supernatant was enzymatically hydrolyzed into glucose by incubating with 320 mM sodium acetate buffer (pH = 4.9) and 40 U·mL$^{-1}$ glucoamylase from *Rhizopus* sp. (TOYOBO, Osaka, Japan) under 200 rpm agitation at 50 °C for 2 h, which was followed by inactivation of the enzyme at 95 °C for 20 min. After centrifugation at $14,000 \times g$ for 5 min, the glucose released into the supernatant was analyzed using a high-performance liquid chromatography system (Shimadzu) equipped with an Aminex HPX-87H column (Bio-Rad Laboratories, Hercules, CA, USA). Glycogen (Nacalai Tesque, Kyoto, Japan) was used as a quantitative standard to determine the glycogen content using a calibration curve[32].

## Metabolome analysis

To analyze the extracellularly released metabolites, the cells in the culture broth were completely removed by centrifugation at $8000 \times g$ for 5 min. The clear supernatant without cells (500 μL) was mixed with 500 μL chloroform that was pre-cooled at 4 °C. After centrifugation at $14,000 \times g$ for 5 min at 4 °C, the upper layer was collected and filtered using UFC5003BK (Merck Millipore). The flow through was added with 400 μM L-methionine sulfone and 400 μM piperazine-1,4-bis(2-ethanesulfonic acid) (PIPES) as the internal standards[29]. To prepare intracellular metabolites, cells equivalent to 5 mg dry weight were harvested from the culture broth by filtration using 1 μm pore size polytetrafluoroethylene filters (Merck Millipore). The cells on the filter were washed once with 20 mM ammonium carbonate (pre-cooled at 4 °C) and immediately resuspended in 2 mL of methanol (pre-cooled at −30 °C) containing 37.3 μM L-methionine sulfone and 37.3 μM PIPES as internal standards. The cell suspension (500 μL) was added to 200 μL ultrapure water and 500 μL chloroform pre-cooled at 4 °C and then vigorously mixed using vortexing for 30 s. After centrifugation at $14,000 \times g$ for 5 min at 4 °C, the aqueous layer was collected and filtered through an Amicon Ultra-0.5 Centrifugal Filter Unit UFC5003BK (Merck Millipore) by centrifugation at $14,000 \times g$ at 4 °C. The sample (300 μL) was dried under vacuum using a centrifugal evaporator CEV-3100 (EYELA, Tokyo, Japan) and resuspended in 20 μL of ultrapure water[33]. Extracellular and intracellular samples were subjected to capillary electrophoresis time-of-flight mass spectrometry (CE-TOF MS) using a G7100 CE and G6224AA liquid chromatography-mass selective detector (LC/MSD) TOF system (Agilent Technologies, Santa Clara, CA, USA).

To investigate the de novo synthesis of metabolites, the cell culture on day 5 was subjected to $^{13}C$ labeling. To label newly synthesized metabolites from $CO_2$, the $K_2CO_3/KHCO_3$ solution in the lower stage of the double-deck flasks was removed, and 25 mM $NaH^{13}CO_3$ was added to the cell culture as a carbon source. After 0–12 h of cultivation under the phototrophic conditions described above, intracellular metabolites were prepared and subjected to CE-TOF MS analysis. The ratio of $^{13}C$ to total carbon ($^{13}C$ fraction) of the metabolite was determined based on shifts between the $^{12}C$ and $^{13}C$ mass spectra[29].

## Statistics and reproducibility

Data are presented as the mean ± standard deviation of three replicate experiments. Statistical significance was determined using the Welch's *t* test.

## Reporting summary

Further information on research design is available in the Nature Portfolio Reporting Summary linked to this article.

## Data availability

The source data underlying Figs. 1–4 are provided in Supplementary Data 1. The data supporting the findings of this study are available from the corresponding author upon request.

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

## Acknowledgements
The authors thank Ms. Hiroko Koizumi for their technical assistance. This work was supported by the Japan Science and Technology Agency (JST), Mirai Program Grant Number JPMJMI19E4, and the Ministry of Education, Culture, Sports, Science, and Technology (MEXT), Japan.

## Author contributions
Y.K. designed the study, conducted experiments, and drafted the paper. R.H. interpreted the results and revised the paper. M. M. conducted the experiments. R.O. and H.A. interpreted the results. A.K. commented on the study and assisted with laboratory management. T.H. designed the study, revised the paper, and supervised the study. All the authors have read and approved the final version of this paper.

## Competing interests
The authors declare no competing interests.
