## [Peer Review File · Communications Biology]

Reviewers' comments:

Reviewer #1 (Remarks to the Author):

Reviewer report

In Kato et al., authors analyzed changes in carbon partitioning of a *glgC* knockout strain of *Synechococcus elongatus* PCC 7942, unable to synthesize and accumulate glycogen because of the absence of the first enzyme in the pathway of glycogen synthesis. Extracellular and intracellular levels of metabolites were measured under two nitrogen concentrations supplemented in the media (low level/7.5 mM NaNO₃ and high level/17.6 mM NaNO₃). In addition, cultures of wild type and Δ *glgC* mutant were labeled with NaH¹³CO₃ to determine the de novo synthesis of key metabolites in the glycolysis, the tricarboxylic acid (TCA) cycle and the glutamine synthetase–glutamate synthase (GS–GOGAT) cycle. These results revealed that in the presence of low levels of nitrogen the Δ *glgC* mutant is unable to accumulate glutamate, and large levels of 2-oxoglutarate and pyruvate are extracellularly detected. By contrary, under high nitrogen conditions only glutamate is secreted to the media in the Δ *glgC* mutant, by an unidentified transport mechanism, although authors hypothesized that a mechanosensitive channel can be involved as previously reported for other bacteria. In absence of the endogenous glycogen sink, glutamate serves as an alternative extracellular sink under HN conditions.

The results presented in this manuscript are solid and would provide useful information to the researchers working in related fields. However, I have some major comments and questions that can contribute to a better and clearer manuscript.

Major comments

- 1) Along the manuscript, the authors interchange the nomenclature of the conditions used in this study by referring to low nitrogen (LN) as nitrogen-depleted conditions and high nitrogen (HN) as nitrogen-repleted conditions. Although, it is true that the nitrogen supplied under LN conditions is rapidly depleted (as shown in Figure 1), it takes at least 3 days to get to that point. The changes observed between 0 and 3 days are not due to nitrogen depletion since the disappearance of this substrate does not occur immediately. Clarification of these differences with a nitrogen-depleted situation should be discussed in the manuscript, as well as changing the terminology referring the results to LN and HN, rather to talk about nitrogen-depleted and nitrogen-repleted conditions, respectively.
- 2) The authors refer to the *glgC* knockout mutant as *glgC* mutant, an incomplete nomenclature of this strain. Strain name should be changed to “*glgC* knockout mutant” or “ Δ *glgC* mutant” to be more specific with the kind of mutant used in this study. Please, make sure that you make the appropriate changes to the document.
- 3) The paragraphs in the introduction are not connected between them. The last sentence of each paragraph should have a small “hint” of what the next paragraph is going to be about. For example, the last sentence of first paragraph should be focused on the fact that glycogen seems to be dispensable in continuous light conditions, but it is essential in diel cycles, and more specifically, in the dark phase. This main idea connects with the thought that the pathway can be removed to increase production of certain compounds of industrial interest. In addition, the last sentence of the second paragraph should mention that it has been previously reported that the modification of glycogen pathways has been associated with releasing metabolites to the media (this is the core idea of paragraph 3).

4) The last paragraph of the introduction includes too many details. The last paragraph of the introduction should forecast your main arguments and conclusions and provide a description of the rest of the paper that shows the reader where you are going and what to expect. Please, reorganize and remove unnecessary details (e.g., the concentrations of nitrogen used – saved it for results and material and methods), and try to combine sentences to make this paragraph shorter and concise.

5) Lines 137-142 – While it is true that pyruvate and 2-oxoglutarate have been previously reported to be extracellularly detected in Δ glgC mutants of *S. elongatus* PCC 7942 (ref. 9, 16), none of the other metabolites have been mentioned previously in literature for this strain. Succinate has been previously detected in the media of Δ glgC mutants of *Synechococcus* sp. PCC 7002 (ref. 18). In addition, pyruvate and 2-oxoglutarate have been secreted by Δ glgC mutants of *Synechocystis* sp. PCC 6803 (ref. 8, 17 and Carrieri et al., 2015 – PMID: 25616027), indicating that this phenomenon is not strain-specific. In any of the Δ glgC mutants of these different cyanobacterial species, lactate, fumarate, and malate have been detected. Please, rephrase this section in the document to include this information that is important for the conclusions of this manuscript.

6) In Table 1, the authors observed statistical differences for other than the ones discussed in the document and shown in the summary figure (Figure 4). For example, glutamate seems to be secreted under LN conditions in comparison to the WT in the same conditions (other examples, are lactate and 2-oxoglutarate). What was the criteria chosen to decide with metabolites are selectively secreted under LN vs HN conditions? Using the fold-change of the metabolite levels in Δ glgC vs WT in each nitrogen condition would be a quantitative way to determine these changes without being influenced by the individual concentrations.

7) A more meritorious discussion of the results shown in Figure 2 is needed. For example, what is the possible hypothesis of the changes observed in ADP, fructose 6-phosphate and 3-PGA levels? Changes in fructose 6-phosphate and 3-PGA levels have been reported in *Synechocystis* sp. PCC 6803 when PHB sink was eliminated under nitrogen-depleted conditions (Hauf et al., 2013 – PMID: PMC3901256). This reference and the possible similarities between both situations should be included in the manuscript.

8) Lines 264-268 – Authors mentioned that mechanosensitive channels might be exporting glutamate in *S. elongatus* PCC 7942 under nitrogen-repleted conditions. Does *S. elongatus* PCC 7942 have homologs of these mechanosensitive channels? If so, which ones and what is the percentage of homology with respect to *Escherichia coli* and *Corynebacterium glutamicum* ones?

Minor points

- Line 26 – Authors should refer to “glycogen-accumulation deficiency” as “glycogen deficiency”, since glycogen is not synthesized, rather than not being supplied.
- Line 34 – “de novo” should be written in italics.
- Lines 36 and 38 – The verb “suggest” is in two consecutive sentences. Please, rephrase one of the sentences to avoid it.
- Lines 38-39 – The conclusion is vague since glutamate is the alternative sink only in HN conditions. Please, rephrase.
- Line 43 – Change “is” for “serve as”.
- Lines 44-47 – Please, restructure this sentence to indicate that glycogen synthesis and accumulation is strongly induced by “unfavorable conditions, including” and cite the examples.
- Lines 51-52 – Indicate the acronyms that are going to be used for the name of each strain in the

manuscript (i.e., PCC7942 for *S. elongatus* PCC 7942, PCC7002 for *Synechococcus* sp. PCC 7942 and PCC6803 for *Synechocystis* sp. PCC 6803).

- Lines 61-63 – Are you trying to say that “blocking of glycogen accumulation is a potential target that allows to mobilize and reallocate carbon to desired compounds”? If so, please rephrase the sentence to state it.

- Lines 89-90 – Please, rephrase the last sentence of this paragraph since the message is unclear.

- Lines 167-169 – This statement is not entirely true since authors are comparing LN conditions vs nitrogen-depleted conditions in ref. 9. Please, rephrase the sentence to include these changes.

- Line 195 – “As Pyr, 2-OG, and Glu were highly released extracellularly”. What does it mean “highly released”? It is unclear which criteria authors used for deciding high or low release levels of each metabolite (see comment above). Please, rephrase.

- Lines 242-244, 244-249 and 301-303 – Please, rephrase these sentences since it is unclear what authors were trying to say here.

- Line 284 – “because of the low ADP levels”. As mentioned above, this result was not explained before.

- Lines 295-297 – A reference is missing in this sentence.

- Line 316 – Change the units of NaNO₃ to mM instead of M since it is the unit shown in the rest of the document.

Figures and tables

- Figure 1 – The title of this figure is incomplete. Adding subpanels and the corresponding references of these subpanels in the text will make it easier to connect the results with the data. Why are not data points for nitrate concentrations in LN conditions after 6 days? Even if the values are low, the data should be shown since the measurement of metabolite levels was performed after 6 days, more specifically at day 10.

- Table 1 – Please, transfer the data to a bar graph (or even two, one for the metabolites of glycolysis and TCA cycle, and the other one for proteinogenic amino acids) and move the data from Table 1 to a supplementary table. There are so many numbers to compare, and a visual graph will help to make it easier to make the comparisons between both strains and both conditions. Also, please include in the title of the supplementary table/new figure that the values were measured at day 10, since it will help to connect the information with figure 1. Do you have data for the levels of these metabolites at other time points? If so, please include them in the supplementary material.

- Figure 2 – This figure should be split into two figures, for example, one figure for TCA cycle metabolites and another one for CBB cycle metabolites. The panels are so small that make it harder for the reader to see the values and changes. I would also suggest including the values of all the metabolites levels in this graph in a supplementary table. Also, why were lactate levels not measured in this experiment?

- Figure 3 – A schematic diagram of the pathways and metabolites analyzed by ¹³C labelling experiments will be useful to understand better the results shown in this figure and indicating the points of entry of NaH¹³CO₃ in the metabolic pathways of *S. elongatus* PCC 7942.

- Figure 4 – As mentioned in one of the comments above, it would be more informative to include which other metabolites (other minor sinks) that are acting under each nitrogen condition (data from Table 1) using difference thickness of the arrows.

Reviewer #2 (Remarks to the Author):

In the manuscript under consideration the authors identify glutamate as the main alternative metabolic sink in the absence of glycogen. The analysis of metabolites excreted into the culture medium reveals that the glycogen-deficient mutant strain excretes glutamate under nitrogen-rich conditions. However, no intracellular accumulation of glutamate is detected under this condition, leading the authors to propose an active glutamate export mechanism, which could serve as a metabolic and energy sink in the absence of glycogen synthesis. The results also confirm the pyruvate and 2-oxoglutarate intracellular accumulation and excretion to the medium in the glycogen-less mutant under nitrogen-deficient conditions, as previously described in cyanobacteria.

In my opinion, the manuscript presents interesting original results and provide novel findings employing a straightforward experimental design. The paper is concise and well-written making it easy to read and follow. In addition, methods are described clearly and in detail. However, I have the following concerns and comments:

1. The whole story relies on the detection of glutamate in the measurement of extracellular metabolites of the glgC mutant. The sampling process for extracellular metabolites indicated in the Materials and methods section involves centrifugation and direct mixing of the supernatant with chloroform. Would this release the cellular contents of any potential remaining cells? Have any controls been conducted to confirm that the presence of metabolites in the culture medium is not due to the presence of unpelleted cells following centrifugation or to a potential cell rupture in the mutant strain? It would have been prudent to consider filtration of the supernatant before mixing with chloroform to remove any remaining cells. Special care must be taken in this point, since glutamate is a highly abundant metabolite in the cell. In fact, according to the data presented in this work, it is the most abundant of the metabolites measured intracellularly (up to almost 80000 nmol/mg DCW), almost an order of magnitude higher than the second most abundant measured metabolite, 2-oxoglutarate (that reaches 10000 nmol/mg DCW in the glgC under nitrogen deprivation)
2. Additionally, data calculation and/or units of metabolite concentration should be carefully revised. In Fig.2 intracellular metabolites concentration is expressed in nmol/mg CDW. A concentration of aprox. 70000 nmol/mg CDW is indicated for glutamate at time 5 days in high nitrogen. This corresponds to a glutamate concentration of 10 mg/mg CDW, which makes no sense. Maybe is just a typo between nmol/mg CDW and nmol/g CDW, but calculations should be revised.
3. The detection of elevated glutamate levels in the culture medium of the mutant strain, despite the absence of intracellular accumulation, leads the authors to propose the existence of a selective export mechanism, similar to those described in other organisms. This unidentified mechanism would be activated in response to glycogen deficiency. It would be valuable to expand on this point and explore whether homologous proteins exist in *Synechococcus elongatus* PCC 7942 and whether similar systems have been described in other cyanobacteria in the literature.

Minor comments:

1. Line 51-52: Specify that henceforth, strains will be referred to as PCC6803, etc.

2. Line 122 and Fig. 1: Glycogen content is expressed just as a percentage. I assume is a % relative to CDW, but it should be stated in the graph axis and text.
3. Line 138 and Table 1: For me, it is quite surprising the presence of lactate in the medium. To my knowledge, wild type *Synechococcus* PCC 7942 doesn't excrete lactate. Lactate excretion has been reported in genetically modified strains expressing a heterologous set of enzymes (e.g. in Niederholtmeyer H, et al. Engineering cyanobacteria to synthesize and export hydrophilic products. *Appl Environ Microbiol.* 2010. doi: 10.1128/AEM.00202-10; where expression of *E. coli* lactate dehydrogenase and lactate transporter yields extracellular lactate levels of 300 μM (aprox. 20 mg/L) after 9 days. This value is similar to level of lactate measured in this work for the *glgC* mutant under nitrogen deprivation (aprox. 15 mg/L). Is there any reference of lactate production for *Synechococcus* strains not genetically modified for this purpose? (Not questioning the data, just curious)
4. Line 157 and following: Why haven't the values at the 0-hour time point of the experiment been included?
5. Line 293 and following: The authors suggest that Glu export into the medium might be mediated by a mechanosensitive exporter in response to hypoosmotic stress in the *glgC* mutant. It would be nice to investigate whether different osmotic concentrations in the culture medium affect the levels of Glu detected extracellularly.

Reviewer #3 (Remarks to the Author):

Review on manuscript COMMSBIO-23-3236 "Glycogen deficiency enhances carbon partitioning into glutamate for an alternative extracellular metabolic sink in cyanobacteria" and is authored by Professor Hasunuma and colleagues.

The use of cyanobacteria and microalgae is often discussed to provide an opportunity for CO₂ neutral green production of fuels and feedstock. Therefore, a better understanding of metabolic regulation of carbon flow is mandatory. In this regard, deletion of the glycogen sink is an attractive option to reduce carbon partitioning in this sink. Previous studies showed that glycogen less cyanobacterial cells release pyruvate and 2OG under N-limited conditions into the medium. This overflow mechanism is believed to protect the cell for photoinhibition. In the present study the authors basically showed that under high CO₂ conditions in the presence of available nitrate glutamate is the preferred extracellular compound. This finding is new and interesting. However, it is not surprising that glutamate is released instead of 2OG when nitrate can be used to produce ammonia for the GS/GOGAT cycle. Such conditions were not investigated before.

I would like to suggest some points that might help to improve the manuscript:

1. Line 39: probably intracellular (only when this pool is filled then it is released!)
2. Introduction: Orthwein et al. (2021) identified the PirC protein that controls the carbon-partitioning between glycogen and TCA sinks under N-limiting conditions, this paper and the therein described mechanism should be considered by the authors!

3. Line 102: I did not find any evidence for the conversion of Glu into Pyr and 2OG. It is more likely that ammonia assimilation drops and therefore less glutamate is made and then instead of Glu the precursors pyruvate and 2OG are released!
4. Line 110: please specify, what is the standard nitrate amount in BG11
5. Line 111: please add: "under high CO₂ conditions (1%)". This information is important for the reader, because at ambient air the overall flux into glycogen is very low and probably different results would have been obtained.
6. Line 112: Fig. 1 upper left panel indicates significantly more nitrate in cultures with mutant cells at day 2, how can then the nitrate consumption rate be similar?
7. Line 125: maybe also the release of metabolites contribute to the slower growth? because a high proportion of carbon and nitrogen is released. Please quantify the carbon loss to the relative biomass increase.
8. Line 136: unfortunately, only day 10 was analyzed, when both cultures are already N-limited, a second time point after day 4 would be highly interesting, when HN cultures still have a good N supply. Intracellular metabolite data for this time point are available, why not for the medium?
9. Line 144: How can you rule out that cell lysis instead of metabolite release from viable cells contribute to the observed values? It would be good to see that cells of both strains are viable after 10 days. Glu is by far the most abundant amino acid, hence, cell lysis would easily explain its dominance in the medium.
10. Line 163: ADP levels decrease much more than ATP in the mutant, therefore, the energy charge ATP/ADP is better in mutant cells and should be not limit Calvin cycle.
11. Line 172: Glu was similar only at day 2, afterwards lower Glu levels in cells under LN conditions were observed, even in mutant under HN conditions. Interestingly, also Gln contents are much lower after day 2 in glgC mutant at LN, indicating the expected strong decrease of ammonia assimilation due to missing nitrate source and thereby decreased GS/GOGAT products.
12. Line 206: this I cannot see, the ¹³C enrichment into 3PGA is similar over 12 h in glgC and WT under LN conditions (no nitrate at day 5), while it is lower mutant than WT grown for 5 days in HN (nitrate available), hence nitrogen replete conditions (HN) decrease 3PGA synthesis and not nitrogen-deplete conditions (LN).
13. Line 220: Difference in Glu labelling much more pronounced in LN cells not releasing Glu, when no nitrate is available
14. Line 214. This finding is surprising, because at least under LN conditions almost no ammonia should be available for new assimilation, probably, these increases are due to the different pool sizes, the authors should normalize the ¹³C enrichment by pool size, then the real synthesis rate is visible.
15. Line 220: Glu only released from HN cells, but Glu enrichment much higher in LN cells, this easy conclusion is not really supported by the data
16. Line 241/242: This statement is not true. The same carbon partitioning occurs into pyruvate, 2OG and Glu occurs (export of C into lower glycolysis and TCA cycle), the previous studies have been done mostly under low CO₂ conditions and complete absence of N source. Therefore, the Glu excretion was overlooked before.
17. Line 244: it's not only due to the nitrate but also because of high CO₂!
18. Line 251: good hypothesis, why not measuring residual P in the medium from HN cultures.
19. Line 266: mechanosensitive channels are known to release low molecular mass substances without

selectivity. Hence, the relative proportion of extracellular substances should be similar to the intracellular composition. Please compare this for HN cells to support or disprove the suggestion

20. Line 289: this is true, but according to your model in Fig. 4, Glu can be only made when ammonia is produced from nitrate, i.e. under HN conditions in the present study!

21. Line 296: I think the authors mean hyperosmotic

KOBE UNIVERSITY

Tomohisa Hasunuma
Engineering Biology Research Center
1-1 Rokkodai, Nada, Kobe 657-8501, Japan
Tel. +81-78-803-6202 Fax. +81-78-803-6102
E-mail: hasunuma@port.kobe-u.ac.jp

Responses to comments

We thank the editor and reviewers for the effort and time spent in carefully reviewing our manuscript and for the helpful comments and suggestions. We thoroughly revised the manuscript based on the comments, and as a result, the manuscript has been improved dramatically. Our point-by-point responses to the comments are provided below. The changes in the revised manuscript are shown in **red text**.

Reviewer #1 (Remarks to the Author):

Reviewer report

In Kato et al., authors analyzed changes in carbon partitioning of a glgC knockout strain of Synechococcus elongatus PCC 7942, unable to synthesize and accumulate glycogen because of the absence of the first enzyme in the pathway of glycogen synthesis. Extracellular and intracellular levels of metabolites were measured under two nitrogen concentrations supplemented in the media (low level/7.5 mM NaNO₃ and high level/17.6 mM NaNO₃). In addition, cultures of wild type and ΔglgC mutant were labeled with NaH¹³CO₃ to determine the de novo synthesis of key metabolites in the glycolysis, the tricarboxylic acid (TCA) cycle and the glutamine synthetase–glutamate synthase (GS–GOGAT) cycle. These results revealed that in the presence of low levels of nitrogen the ΔglgC mutant is unable to accumulate glutamate, and large levels of 2-oxoglutarate and pyruvate are extracellularly detected. By contrary, under high nitrogen conditions only glutamate is secreted to the media in the ΔglgC mutant, by an unidentified transport mechanism, although authors hypothesized that a mechanosensitive channel can be involved as previously reported for other bacteria. In absence of the endogenous glycogen sink, glutamate serves as an alternative extracellular sink under HN conditions. The results presented in this manuscript are solid and would provide useful information to the researchers working in related fields. However, I have some major comments and questions that can contribute to a better and clearer manuscript.

Major comments

1) Along the manuscript, the authors interchange the nomenclature of the conditions used in this study by referring to low nitrogen (LN) as nitrogen-depleted conditions and high nitrogen (HN) as nitrogen-repleted conditions. Although, it is true that the nitrogen supplied under LN conditions is rapidly depleted (as shown in Figure 1), it takes at least

KOBE UNIVERSITY

Tomohisa Hasunuma
Engineering Biology Research Center
1-1 Rokkodai, Nada, Kobe 657-8501, Japan
Tel. +81-78-803-6202 Fax. +81-78-803-6102
E-mail: hasunuma@port.kobe-u.ac.jp

3 days to get to that point. The changes observed between 0 and 3 days are not due to nitrogen depletion since the disappearance of this substrate does not occur immediately. Clarification of these differences with a nitrogen-depleted situation should be discussed in the manuscript, as well as changing the terminology referring the results to LN and HN, rather to talk about nitrogen-depleted and nitrogen-repleted conditions, respectively.

Response:

We thank the reviewer for this important comment. By adding details in the Results (Line 110–113) and Discussion section (Line 270–277), we make it obvious that, under LN conditions, the wild-type and $\Delta glgC$ strains might be considered nitrogen depleted after day 5. In addition, we carefully checked and revised the manuscript to avoid simply referring to the LN and HN conditions as nitrogen-depleted and nitrogen-repleted conditions.

2) The authors refer to the $glgC$ knockout mutant as $glgC$ mutant, an incomplete nomenclature of this strain. Strain name should be changed to “ $glgC$ knockout mutant” or “ $\Delta glgC$ mutant” to be more specific with the kind of mutant used in this study. Please, make sure that you make the appropriate changes to the document.

Response:

We thank the reviewer for this valuable comment. According to the reviewer's suggestion, we changed the nomenclature “ $glgC$ mutant” to “ $\Delta glgC$ mutant”.

3) The paragraphs in the introduction are not connected between them. The last sentence of each paragraph should have a small “hint” of what the next paragraph is going to be about. For example, the last sentence of first paragraph should be focused on the fact that glycogen seems to be dispensable in continuous light conditions, but it is essential in diel cycles, and more specifically, in the dark phase. This main idea connects with the thought that the pathway can be removed to increase production of certain compounds of industrial interest. In addition, the last sentence of the second paragraph should mention that it has been previously reported that the modification of glycogen pathways has been associated with releasing metabolites to the media (this is the core idea of paragraph 3).

Response:

KOBE UNIVERSITY

Tomohisa Hasunuma
Engineering Biology Research Center
1-1 Rokkodai, Nada, Kobe 657-8501, Japan
Tel. +81-78-803-6202 Fax. +81-78-803-6102
E-mail: hasunuma@port.kobe-u.ac.jp

We thank the reviewer for this comment. According to the reviewer's suggestion, the last sentence of the first paragraph was revised (Line 56–57). In addition, we added a sentence at the last of the second paragraph to connect it to the next paragraph (Line 68–69).

4) The last paragraph of the introduction includes too many details. The last paragraph of the introduction should forecast your main arguments and conclusions and provide a description of the rest of the paper that shows the reader where you are going and what to expect. Please, reorganize and remove unnecessary details (e.g., the concentrations of nitrogen used – saved it for results and material and methods), and try to combine sentences to make this paragraph shorter and concise.

Response:

We thank the reviewer for this comment. We have reorganized the last paragraph of the introduction and made it shorter by removing detailed descriptions such as nitrogen concentration.

*5) Lines 137-142 – While it is true that pyruvate and 2-oxoglurate have been previously reported to be extracellularly detected in Δ glgC mutants of *S. elongatus* PCC 7942 (ref. 9, 16), none of the other metabolites have been mentioned previously in literature for this strain. Succinate has been previously detected in the media of Δ glgC mutants of *Synechococcus* sp. PCC 7002 (ref. 18). In addition, pyruvate and 2-oxoglurate have been secreted by Δ glgC mutants of *Synechocystis* sp. PCC 6803 (ref. 8, 17 and Carrieri et al., 2015 – PMID: 25616027), indicating that is this phenomenon is not strain-specific. In any of the Δ glgC mutants of these different cyanobacterial species, lactate, fumarate, and malate have been detected. Please, rephrase this section in the document to include this information that is important for the conclusions of this manuscript.*

Response:

We thank the reviewer for indicating this point. We rephrased this section and added detailed descriptions of released metabolites in several cyanobacteria (Line 142–146).

6) In Table 1, the authors observed statistical differences for other than the ones discussed in the document and shown in the summary figure (Figure 4). For example, glutamate seems to be secreted under LN conditions in comparison to the WT in the same conditions

KOBE UNIVERSITY

Tomohisa Hasunuma
Engineering Biology Research Center
1-1 Rokkodai, Nada, Kobe 657-8501, Japan
Tel. +81-78-803-6202 Fax. +81-78-803-6102
E-mail: hasunuma@port.kobe-u.ac.jp

(other examples, are lactate and 2-oxoglutarate). What was the criteria chosen to decide with metabolites are selectively secreted under LN vs HN conditions? Using the fold-change of the metabolite levels in Δ glgC vs WT in each nitrogen condition would be a quantitative way to determine these changes without being influenced by the individual concentrations.

Response:

We thank the reviewer for this comment. Although we do not have any general criteria currently, we have revised to show a concrete indicator (more than $10 \text{ mg}\cdot\text{L}^{-1}$ in the medium at day 10) as a "highly" released metabolite in the manuscript (Line 140). Please note that we have revised the manuscript to show time course changes of extracellular metabolites as Figure 2, according to the comments below. The data of extracellular metabolites at day 10, previously shown in Table 1, has now been shown in Supplementary Table 1.

*7) A more meritorious discussion of the results shown in Figure 2 is needed. For example, what is the possible hypothesis of the changes observed in ADP, fructose 6-phosphate and 3-PGA levels? Changes in fructose 6-phosphate and 3-PGA levels have been reported in *Synechocystis* sp. PCC 6803 when PHB sink was eliminated under nitrogen-depleted conditions (Hauf et al., 2013 – PMID: PMC3901256). This reference and the possible similarities between both situations should be included in the manuscript.*

Response:

We thank the reviewer for this comment regarding the figure (currently shown as Figure 3). Unfortunately, we do not have any confident hypothesis for the changes in these metabolites, we just added a simple suggestion based on the 3-PGA levels in the Δ glgC mutant (Line 177–178).

We also thank the reviewer for showing us the paper by Hauf *et al.* (2013) analyzing metabolites in the low PHB mutant of PCC 6803 upon nitrogen depletion. We added a discussion that compares levels of the intracellular metabolites in the Δ glgC mutant with those in the low PHB mutant (Line 287–295).

*8) Lines 264-268 – Authors mentioned that mechanosensitive channels might be exporting glutamate in *S. elongatus* PCC 7942 under nitrogen-repleted conditions. Does*

KOBE UNIVERSITY

Tomohisa Hasunuma
Engineering Biology Research Center
1-1 Rokkodai, Nada, Kobe 657-8501, Japan
Tel. +81-78-803-6202 Fax. +81-78-803-6102
E-mail: hasunuma@port.kobe-u.ac.jp

S. elongatus PCC 7942 have homologs of these mechanosensitive channels? If so, which ones and what is the percentage of homology with respect to *Escherichia coli* and *Corynebacterium glutamicum* ones?

Response:

We thank the reviewer for this comment. PCC 7942 has 3 possible mechanosensitive channel genes. The large conductance mechanosensitive channel protein MscL (Synpcc7942_1991) is 40.2% identical to the MscL protein in *E. coli* and 37.5% identical to the MscL in *C. glutamicum*. The mechanosensitive ion channel family protein (Synpcc7942_0610) is 28.7% identical to the moderate conductance mechanosensitive channel protein YbiO in *E. coli* and 31.4% identical to the mechanosensitive ion channel family protein in *C. glutamicum* (WP_077312967). The mechanosensitive ion channel protein (Synpcc7942_0664) is 33.6%, 33.3%, and 29.1% identical to the potassium-dependent small conductance mechanosensitive channel MscK, mini conductance mechanosensitive channel MscM, and small conductance mechanosensitive channel MscS in *E. coli*, respectively, and 23.7% identical to the CRP-like cAMP-activated global transcriptional regulator GlxR in *C. glutamicum* (WP_006285706). We added a description of these mechanosensitive channel genes harbored by PCC 7942 in the discussion (Line 314–319).

Minor points

- Line 26 – Authors should refer to “glycogen-accumulation deficiency” as “glycogen deficiency”, since glycogen is not synthesized, rather than not being supplied.

Response:

We thank the reviewer for this comment. We have referred to the phenotype as “glycogen deficiency” (Line 26).

- Line 34 – “de novo” should be written in italics.

Response:

We thank the reviewer for this comment. All “de novo” were revised to be written in italics.

KOBE UNIVERSITY

Tomohisa Hasunuma
Engineering Biology Research Center
1-1 Rokkodai, Nada, Kobe 657-8501, Japan
Tel. +81-78-803-6202 Fax. +81-78-803-6102
E-mail: hasunuma@port.kobe-u.ac.jp

- Lines 36 and 38 – The verb “suggest” is in two consecutive sentences. Please, rephrase one of the sentences to avoid it.

Response:

We thank the reviewer for this comment. The second description was rephrased as “This study proposes a model in which ...” (Line 36–37).

- Lines 38-39 – The conclusion is vague since glutamate is the alternative sink only in HN conditions. Please, rephrase.

Response:

We thank the reviewer for this comment. We added “when nitrogen is available” at the last of the sentence for more definite writing (Line 38).

- Line 43 – Change “is” for “serve as”.

Response:

We thank the reviewer for this comment. We changed the description to “serves as” (Line 41).

- Lines 44-47 – Please, restructure this sentence to indicate that glycogen synthesis and accumulation is strongly induced by “unfavorable conditions, including” and cite the examples.

Response:

We thank the reviewer for this comment. We restructured the sentence according to the reviewer’s suggestion (Line 43–45).

*- Lines 51-52 – Indicate the acronyms that are going to be used for the name of each strain in the manuscript (i.e., PCC7942 for *S. elongatus* PCC 7942, PCC7002 for *Synechococcus* sp. PCC 7942 and PCC6803 for *Synechocystis* sp. PCC 6803).*

Response:

We thank the reviewer for this comment. We revised the sentence to indicate the acronyms

KOBE UNIVERSITY

Tomohisa Hasunuma
Engineering Biology Research Center
1-1 Rokkodai, Nada, Kobe 657-8501, Japan
Tel. +81-78-803-6202 Fax. +81-78-803-6102
E-mail: hasunuma@port.kobe-u.ac.jp

for these cyanobacteria (Line 50–51).

- Lines 61-63 – Are you trying to say that “blocking of glycogen accumulation is a potential target that allows to mobilize and reallocate carbon to desired compounds”? If so, please rephrase the sentence to state it.

Response:

We thank the reviewer for this comment. Due to potential readability difficulties, we have rephrased the sentence (Line 60–61).

- Lines 89-90 – Please, rephrase the last sentence of this paragraph since the message is unclear.

Response:

We thank the reviewer for this comment. I rephrased it to make the message clear (Line 89–90).

- Lines 167-169 – This statement is not entirely true since authors are comparing LN conditions vs nitrogen-depleted conditions in ref. 9. Please, rephrase the sentence to include these changes.

Response:

We thank the reviewer for this comment. We rephrase the sentence (Line 188–192).

- Line 195 – “As Pyr, 2-OG, and Glu were highly released extracellularly”. What does it mean “highly released”? It is unclear which criteria authors used for deciding high or low release levels of each metabolite (see comment above). Please, rephrase.

Response:

We thank the reviewer for this comment. We rephrased the sentence (Line 223–224).

- Lines 242-244, 244-249 and 301-303 – Please, rephrase these sentences since it is unclear what authors were trying to say here.

KOBE UNIVERSITY

Tomohisa Hasunuma
Engineering Biology Research Center
1-1 Rokkodai, Nada, Kobe 657-8501, Japan
Tel. +81-78-803-6202 Fax. +81-78-803-6102
E-mail: hasunuma@port.kobe-u.ac.jp

Response:

We thank the reviewer for this comment. Because we thoroughly revised this section based on the comments below, the sentences in Lines 242-244 and 244-249 were deleted as a result. Also, the sentence in 301-303 was deleted since it was redundant.

- Line 284 – “because of the low ADP levels”. As mentioned above, this result was not explained before.

Response:

We thank the reviewer for this comment. We revised the manuscript to explain about the low ADP levels in the Result section (Line 180–181).

- Lines 295-297 – A reference is missing in this sentence.

Response:

We thank the reviewer for this comment. References were added to this sentence (Line 355).

- Line 316 – Change the units of NaNO₃ to mM instead of M since it is the unit shown in the rest of the document.

Response:

We thank the reviewer for this comment. We change the units of NaNO₃ in this sentence (Line 374).

Figures and tables

- Figure 1 – The title of this figure is incomplete. Adding subpanels and the corresponding references of these subpanels in the text will make it easier to connect the results with the data. Why are not data points for nitrate concentrations in LN conditions after 6 days? Even if the values are low, the data should be shown since the measurement of metabolite levels was performed after 6 days, more specifically at day 10.

Response:

We thank the reviewer for this comment. We rephrased the title of Figure 1 and added

KOBE UNIVERSITY

Tomohisa Hasunuma
Engineering Biology Research Center
1-1 Rokkodai, Nada, Kobe 657-8501, Japan
Tel. +81-78-803-6202 Fax. +81-78-803-6102
E-mail: hasunuma@port.kobe-u.ac.jp

subpanels to this figure. Data points for nitrate concentrations in LN conditions after 6 days were not shown because we did not measure it after confirming its complete depletion. We think that nitrate concentration in the medium basically does not increase during phototrophic cultivation of cyanobacteria without supplying the other nitrogen source.

- Table 1 – Please, transfer the data to a bar graph (or even two, one for the metabolites of glycolysis and TCA cycle, and the other one for proteinogenic amino acids) and move the data from Table 1 to a supplementary table. There are so many numbers to compare, and a visual graph will help to make it easier to make the comparisons between both strains and both conditions. Also, please include in the title of the supplementary table/new figure that the values were measured at day 10, since it will help to connect the information with figure 1. Do you have data for the levels of these metabolites at other time points? If so, please include them in the supplementary material.

Response:

We thank the reviewer for this valuable comment. We transferred the data (previously shown as Table 1) to graphs including other time points (currently shown in Figure 2). The numerical data were moved to the supplementary information and shown in Supplementary Table 1.

- Figure 2 – This figure should be split into two figures, for example, one figure for TCA cycle metabolites and another one for CBB cycle metabolites. The panels are so small that make it harder for the reader to see the values and changes. I would also suggest including the values of all the metabolites levels in this graph in a supplementary table. Also, why were lactate levels not measured in this experiment?

Response:

We thank the reviewer for this comment. We split the figure previously shown as Figure 2 into two figures (i.e., Figure 3A and 3B). In addition, we added the numerical data to the supplementary information shown in Supplementary Table 2. In addition, we added the data on intracellular lactate levels in Figure 3A and revised the manuscript accordingly (Line 185–189, Line 321–325).

KOBE UNIVERSITY

Tomohisa Hasunuma
Engineering Biology Research Center
1-1 Rokkodai, Nada, Kobe 657-8501, Japan
Tel. +81-78-803-6202 Fax. +81-78-803-6102
E-mail: hasunuma@port.kobe-u.ac.jp

*- Figure 3 – A schematic diagram of the pathways and metabolites analyzed by ^{13}C labelling experiments will be useful to understand better the results shown in this figure and indicating the points of entry of $\text{NaH}^{13}\text{CO}_3$ in the metabolic pathways of *S. elongatus* PCC 7942.*

Response:

We thank the reviewer for this valuable comment. We revised the figure (currently shown as Figure 4) to show the schematic diagram of the pathways and the points of entry of ^{13}C .

- Figure 4 – As mentioned in one of the comments above, it would be more informative to include which other metabolites (other minor sinks) that are acting under each nitrogen condition (data from Table 1) using difference thickness of the arrows.

Response:

We thank the reviewer for this comment. After we examined this suggestion, we decided not to include the minor sinks in this figure because we consider that highlighting the major metabolic sinks in a simple manner would be better.

Reviewer #2 (Remarks to the Author):

In the manuscript under consideration the authors identify glutamate as the main alternative metabolic sink in the absence of glycogen. The analysis of metabolites excreted into the culture medium reveals that the glycogen-deficient mutant strain excretes glutamate under nitrogen-rich conditions. However, no intracellular accumulation of glutamate is detected under this condition, leading the authors to propose an active glutamate export mechanism, which could serve as a metabolic and energy sink in the absence of glycogen synthesis. The results also confirm the pyruvate and 2-oxoglutarate intracellular accumulation and excretion to the medium in the glycogen-less mutant under nitrogen-deficient conditions, as previously described in cyanobacteria.

In my opinion, the manuscript presents interesting original results and provide novel findings employing a straightforward experimental design. The paper is concise and well-written making it easy to read and follow. In addition, methods are described clearly and

KOBE UNIVERSITY

Tomohisa Hasunuma
Engineering Biology Research Center
1-1 Rokkodai, Nada, Kobe 657-8501, Japan
Tel. +81-78-803-6202 Fax. +81-78-803-6102
E-mail: hasunuma@port.kobe-u.ac.jp

in detail. However, I have the following concerns and comments:

*1. The whole story relies on the detection of glutamate in the measurement of extracellular metabolites of the *glgC* mutant. The sampling process for extracellular metabolites indicated in the Materials and methods section involves centrifugation and direct mixing of the supernatant with chloroform. Would this release the cellular contents of any potential remaining cells? Have any controls been conducted to confirm that the presence of metabolites in the culture medium is not due to the presence of unpelleted cells following centrifugation or to a potential cell rupture in the mutant strain? It would have been prudent to consider filtration of the supernatant before mixing with chloroform to remove any remaining cells. Special care must be taken in this point, since glutamate is a highly abundant metabolite in the cell. In fact, according to the data presented in this work, it is the most abundant of the metabolites measured intracellularly (up to almost 80000 nmol/mg DCW), almost an order of magnitude higher than the second most abundant measured metabolite, 2-oxoglutarate (that reaches 10000 nmol/mg DCW in the *glgC* under nitrogen deprivation)*

Response:

We thank the reviewer for this comment. We consider that the supernatant was completely separated from the cells via centrifugation, and unpelleted cells were not present in the supernatant. This was confirmed by the fact that, after the centrifugation, the supernatant was clear and the cells could be visually recognized as green pellets. To make this point clear, the description “supernatant” in the Materials and methods section was revised to “clear supernatant without cells” (Line 431–432).

2. Additionally, data calculation and/or units of metabolite concentration should be carefully revised. In Fig.2 intracellular metabolites concentration is expressed in nmol/mg CDW. A concentration of approx. 70000 nmol/mg CDW is indicated for glutamate at time 5 days in high nitrogen. This corresponds to a glutamate concentration of 10 mg/mg CDW, which makes no sense. Maybe is just a typo between nmol/mg CDW and nmol/g CDW, but calculations should be revised.

Response:

We thank the reviewer for this important comment. As the reviewer indicated, the correct

KOBE UNIVERSITY

Tomohisa Hasunuma
Engineering Biology Research Center
1-1 Rokkodai, Nada, Kobe 657-8501, Japan
Tel. +81-78-803-6202 Fax. +81-78-803-6102
E-mail: hasunuma@port.kobe-u.ac.jp

unit is “nmol·g-DCW⁻¹”, therefore we corrected them in Figure 3 (previously shown as Figure 2).

*3. The detection of elevated glutamate levels in the culture medium of the mutant strain, despite the absence of intracellular accumulation, leads the authors to propose the existence of a selective export mechanism, similar to those described in other organisms. This unidentified mechanism would be activated in response to glycogen deficiency. It would be valuable to expand on this point and explore whether homologous proteins exist in *Synechococcus elongatus* PCC 7942 and whether similar systems have been described in other cyanobacteria in the literature.*

Response:

We thank the reviewer for this comment. We added a description of the possible mechanosensitive channel genes in *Synechococcus elongatus* PCC 7942 and previous reports about mechanosensitive channels in cyanobacteria (Line 314–322).

Minor comments:

1. Line 51-52: Specify that henceforth, strains will be referred to as PCC6803, etc.

Response:

We thank the reviewer for this comment. We revised the manuscript to specify these strain names here (Line 50–51).

2. Line 122 and Fig. 1: Glycogen content is expressed just as a percentage. I assume is a % relative to CDW, but it should be stated in the graph axis and text.

Response:

We thank the reviewer for this comment. We revised the figure and manuscript to show the unit as “% of DCW”.

*3. Line 138 and Table 1: For me, it is quite surprising the presence of lactate in the medium. To my knowledge, wild type *Synechococcus* PCC 7942 doesn't excrete lactate. Lactate excretion has been reported in genetically modified strains expressing a*

KOBE UNIVERSITY

Tomohisa Hasunuma
Engineering Biology Research Center
1-1 Rokkodai, Nada, Kobe 657-8501, Japan
Tel. +81-78-803-6202 Fax. +81-78-803-6102
E-mail: hasunuma@port.kobe-u.ac.jp

heterologous set of enzymes (e.g. in Niederholtmeyer H, et al. Engineering cyanobacteria to synthesize and export hydrophilic products. Appl Environ Microbiol. 2010. doi: 10.1128/AEM.00202-10; where expression of E. coli lactate dehydrogenase and lactate transporter yields extracellular lactate levels of 300 μ M (aprox. 20 mg/L) after 9 days. This value is similar to level of lactate measured in this work for the glgC mutant under nitrogen deprivation (aprox. 15 mg/L). Is there any reference of lactate production for Synechococcus strains not genetically modified for this purpose? (Not questioning the data, just curious)

Response:

We thank the reviewer for this comment. To the best of our knowledge, the present study is the first report that wild-type *Synechococcus elongatus* PCC 7942 excretes lactate, although it harbors a D-lactate dehydrogenase gene (*Synpcc7942_1347*). In *Synechococcus* sp. PCC 7002, the wild-type strain was reported to excrete lactate via autofermentation under dark anoxic conditions (McNeely K., et al. Redirecting reductant flux into hydrogen production via metabolic engineering of fermentative carbon metabolism in a cyanobacterium. Appl Environ Microbiol. 2010. doi: 10.1128/AEM.00862-10).

4. Line 157 and following: Why haven't the values at the 0-hour time point of the experiment been included?

Response:

We thank the reviewer for this comment. The 0-hour time point of the experiment is the end of pre-culture of which the initial density and initial status of the cells were not unified. Therefore, we think that we cannot compare these strains strictly at the 0-hour time point and therefore the values were excluded from the figure.

5. Line 293 and following: The authors suggest that Glu export into the medium might be mediated by a mechanosensitive exporter in response to hypoosmotic stress in the glgC mutant. It would be nice to investigate whether different osmotic concentrations in the culture medium affect the levels of Glu detected extracellularly.

Response:

KOBE UNIVERSITY

Tomohisa Hasunuma
Engineering Biology Research Center
1-1 Rokkodai, Nada, Kobe 657-8501, Japan
Tel. +81-78-803-6202 Fax. +81-78-803-6102
E-mail: hasunuma@port.kobe-u.ac.jp

We thank the reviewer for this comment. We are interested in whether different osmotic concentrations in the culture medium affect the extracellular levels of Glu, and we would like to investigate it in future study.

Reviewer #3 (Remarks to the Author):

Review on manuscript COMMSBIO-23-3236 "Glycogen deficiency enhances carbon partitioning into glutamate for an alternative extracellular metabolic sink in cyanobacteria" and is authored by Professor Hasunuma and colleagues.

The use of cyanobacteria and microalgae is often discussed to provide an opportunity for CO₂ neutral green production of fuels and feedstock. Therefore, a better understanding of metabolic regulation of carbon flow is mandatory. In this regard, deletion of the glycogen sink is an attractive option to reduce carbon partitioning in this sink. Previous studies showed that glycogen less cyanobacterial cells release pyruvate and 2OG under N-limited conditions into the medium. This overflow mechanism is believed to protect the cell for photoinhibition. In the present study the authors basically showed that under high CO₂ conditions in the presence of available nitrate glutamate is the preferred extracellular compound. This finding is new and interesting. However, it is not surprising that glutamate is released instead of 2OG when nitrate can be used to produce ammonia for the GS/GOGAT cycle. Such conditions were not investigated before.

I would like to suggest some points that might help to improve the manuscript:

1. Line 39: probably intracellular (only when this pool is filled then it is released!)

Response:

We thank the reviewer for this comment. Glutamate was extracellularly released while the intracellular level was lower in the $\Delta glgC$ mutant than in the wild-type (Figure 4). This result suggested that glutamate was released by a cellular transport mechanism even when the pool was not filled, as we described in the previous sentence in Abstract (Line 36). Therefore, we assume that the “extracellular” transport of glutamate is a part of the metabolic response which utilizes glutamate as the metabolic sink alternative to glycogen.

2. Introduction: Orthwein et al. (2021) identified the PirC protein that controls the

KOBE UNIVERSITY

Tomohisa Hasunuma
Engineering Biology Research Center
1-1 Rokkodai, Nada, Kobe 657-8501, Japan
Tel. +81-78-803-6202 Fax. +81-78-803-6102
E-mail: hasunuma@port.kobe-u.ac.jp

carbon-partitioning between glycogen and TCA sinks under N-limiting conditions, this paper and the therein described mechanism should be considered by the authors!

Response:

We thank the reviewer for this comment. According to the reviewer's comment, we have considered the paper reporting that the PirC protein suppresses carbon flux toward the TCA cycle upon N depletion. As a result, we discovered that this regulation is opposite to what occurred in the $\Delta glgC$ mutant in which synthesis of 2-OG is enhanced (Figure 4 and Supplementary Figure 1). Therefore, we assume that there would be a mechanism in the $\Delta glgC$ mutant different from the regulation of carbon metabolism by the PirC protein, as described in the Discussion section.

3. Line 102: I did not find any evidence for the conversion of Glu into Pyr and 2OG. It is more likely that ammonia assimilation drops and therefore less glutamate is made and then instead of Glu the precursors pyruvate and 2OG are released!

Response:

We apologize for the confusing expression in this sentence. As the reviewer mentioned, we also assume that Pyr and 2-OG are released instead of Glu since ammonia assimilation drops. We have revised the sentence to avoid misunderstanding regarding the release of Pyr and 2-OG (Line 98–100).

4. Line 110: please specify, what is the standard nitrate amount in BG11

Response:

We thank the reviewer for this comment. We have added a description of the standard nitrate amount in BG11 into the sentence (Line 107–108).

5. Line 111: please add: "under high CO₂ conditions (1%)". This information is important for the reader, because at ambient air the overall flux into glycogen is very low and probably different results would have been obtained.

Response:

We thank the reviewer for this comment. Since we would like to focus on nitrogen

KOBE UNIVERSITY

Tomohisa Hasunuma
Engineering Biology Research Center
1-1 Rokkodai, Nada, Kobe 657-8501, Japan
Tel. +81-78-803-6202 Fax. +81-78-803-6102
E-mail: hasunuma@port.kobe-u.ac.jp

condition in this sentence, we referred to CO₂ concentration in the Discussion section (Line 281–282).

6. Line 112: Fig. 1 upper left panel indicates significantly more nitrate in cultures with mutant cells at day 2, how can then the nitrate consumption rate be similar?

Response:

We thank the reviewer for pointing out the incorrect description. We corrected the sentence (Line 108–110).

7. Line 125: maybe also the release of metabolites contribute to the slower growth? because a high proportion of carbon and nitrogen is released. Please quantify the carbon loss to the relative biomass increase.

Response:

We thank the reviewer for this comment. We have revised the sentence that, in addition to the deficiency of glycogen, release of metabolites is also one of the possible reasons for the decreased biomass in the $\Delta glgC$ mutant (Line 127–128). The additionally released metabolites by the $\Delta glgC$ mutant compared to the wild-type at day 10 were approximately 550 mg/L and 120 mg/L under the LN and HN conditions, respectively. In contrast, the decreases of biomass in the $\Delta glgC$ mutant were 1127 mg-DCW/L and 1420 mg/L, under the LN and HN conditions, respectively. Therefore, both glycogen deficit and metabolite release appeared to contribute to slowed growth; however, we believe further research is needed to establish this conclusion.

8. Line 136: unfortunately, only day 10 was analyzed, when both cultures are already N-limited, a second time point after day 4 would be highly interesting, when HN cultures still have a good N supply. Intracellular metabolite data for this time point are available, why not for the medium?

Response:

We thank the reviewer for this comment. We have added the time course data of the extracellular metabolite levels starting from day 3 as Figure 2. The data previously shown in Table 1 has now been shown in Supplementary Table 1.

KOBE UNIVERSITY

Tomohisa Hasunuma
Engineering Biology Research Center
1-1 Rokkodai, Nada, Kobe 657-8501, Japan
Tel. +81-78-803-6202 Fax. +81-78-803-6102
E-mail: hasunuma@port.kobe-u.ac.jp

9. Line 144: How can you rule out that cell lysis instead of metabolite release from viable cells contribute to the observed values? It would be good to see that cells of both strains are viable after 10 days. Glu is by far the most abundant amino acid, hence, cell lysis would easily explain its dominance in the medium.

Response:

We thank the reviewer for this comment. Although we only have preliminary evidence currently, we assume that the cells were not lysed, at least not completely, because we could detect the intracellular metabolites by harvesting the cellular fraction from the culture (Figure 3). In addition, we measured cell number in culture under the HN condition in a preliminary experiment and found that the cell density of these strains did not decrease during the cultivation for 10 days.

10. Line 163: ADP levels decrease much more than ATP in the mutant, therefore, the energy charge ATP/ADP is better in mutant cells and should be not limit Calvin cycle.

Response:

We thank the reviewer for this comment. We corrected the sentence (Line 178–181).

11. Line 172: Glu was similar only at day 2, afterwards lower Glu levels in cells under LN conditions were observed, even in mutant under HN conditions. Interestingly, also Gln contents are much lower after day 2 in glgC mutant at LN, indicating the expected strong decrease of ammonia assimilation due to missing nitrate source and thereby decreased GS/GOGAT products.

Response:

We thank the reviewer for this comment. Missing nitrate source occurred in both strains under the LN conditions, therefore, we assume this is not the cause of lower Glu contents in the *ΔglgC* mutant compared to the wild-type. As we have described in the Discussion session, we hypothesize that the low Glu contents in the *ΔglgC* mutant were caused by its extracellular release via activated mechanosensitive channels (Line 308–314).

12. Line 206: this I cannot see, the ¹³C enrichment into 3PGA is similar over 12 h in

KOBE UNIVERSITY

Tomohisa Hasunuma
Engineering Biology Research Center
1-1 Rokkodai, Nada, Kobe 657-8501, Japan
Tel. +81-78-803-6202 Fax. +81-78-803-6102
E-mail: hasunuma@port.kobe-u.ac.jp

glgC and WT under LN conditions (no nitrate at day 5), while it is lower mutant than WT grown for 5 days in HN (nitrate available), hence nitrogen replete conditions (HN) decrease 3PGA synthesis and not nitrogen-deplete conditions (LN).

Response:

We thank the reviewer for this comment. We have carefully checked the relevant part in the manuscript “Compared to the wild-type, the labeling ratio of 3-PGA in the $\Delta glgC$ mutant remained unchanged and decreased under LN and HN conditions, respectively (Figure 4). This suggests that glycogen deficiency causes a decrease in carbon fixation under nitrogen-replete conditions, but not under nitrogen-depleted conditions.” (Line 231–235), and we think the description matches the reviewer's recognition.

13. Line 220: Difference in Glu labelling much more pronounced in LN cells not releasing Glu, when no nitrate is available

Response:

We thank the reviewer for this comment. As the reviewer indicated, the ^{13}C -labeling “ratio” of Glu was much more pronounced in the LN cells. We assume that the ^{13}C fraction of Glu in the $\Delta glgC$ mutant rapidly increased under LN conditions, possibly because of its low intracellular levels (Line 243–245). When focusing on the ^{13}C -labeled “level” of Glu in the cells, it was higher in the HN cells than in the LN cells. To make this point clear, we added the data of the levels of ^{13}C -labeled metabolites as Supplementary Figure 1. Please also see the below comment.

14. Line 214. This finding is surprising, because at least under LN conditions almost no ammonia should be available for new assimilation, probably, these increases are due to the different pool sizes, the authors should normalize the ^{13}C enrichment by pool size, then the real synthesis rate is visible.

Response:

We thank the reviewer for this comment. We revised the manuscript to show the intracellular levels of 2-OG, Glu, and Gln during the ^{13}C -labeling experiment as Supplementary Figure 1. We have also shown the levels of ^{13}C -labeled metabolites that

KOBE UNIVERSITY

Tomohisa Hasunuma
Engineering Biology Research Center
1-1 Rokkodai, Nada, Kobe 657-8501, Japan
Tel. +81-78-803-6202 Fax. +81-78-803-6102
E-mail: hasunuma@port.kobe-u.ac.jp

were calculated by multiplying the intracellular levels ($\text{nmol}\cdot\text{g}\cdot\text{DCW}^{-1}$) with the ^{13}C fraction (%). These data would be valuable for examining the newly synthesized and intracellularly accumulated levels of these metabolites. Therefore, we revised the manuscript based on the added data (Line 245–251).

15. Line 220: Glu only released from HN cells, but Glu enrichment much higher in LN cells, this easy conclusion is not really supported by the data

Response:

We thank the reviewer for this comment. We revised the sentence (Line 251–253).

16. Line 241/242: This statement is not true. The same carbon partitioning occurs into pyruvate, 2OG and Glu occurs (export of C into lower glycolysis and TCA cycle), the previous studies have been done mostly under low CO₂ conditions and complete absence of N source. Therefore, the Glu excretion was overlooked before.

Response:

We thank the reviewer for this comment. We corrected the description (Line 281–284).

17. Line 244: it's not only due to the nitrate but also because of high CO₂!

Response:

We thank the reviewer for this comment. We revised the manuscript to suggest the possibility that the high CO₂ (1%) conditions in this study might be also the cause of the glycogen accumulation (Line 281–284).

18. Line 251: good hypothesis, why not measuring residual P in the medium from HN cultures.

Response:

We thank the reviewer for this comment. We measured the phosphate concentration in the supernatant and confirmed that phosphorus depletion occurred by day 3. Therefore, we added the data of the phosphate concentration as Figure 1B and revised the Discussion

KOBE UNIVERSITY

Tomohisa Hasunuma
Engineering Biology Research Center
1-1 Rokkodai, Nada, Kobe 657-8501, Japan
Tel. +81-78-803-6202 Fax. +81-78-803-6102
E-mail: hasunuma@port.kobe-u.ac.jp

section accordingly (Line 278–281).

19. Line 266: mechanosensitive channels are known to release low molecular mass substances without selectivity. Hence, the relative proportion of extracellular substances should be similar to the intracellular composition. Please compare this for HN cells to support or disprove the suggestion

Response:

We thank the reviewer for this comment. We have checked the relative proportion of extracellular and intracellular composition of Pyr, 2-OG, and Glu under the HN conditions. Unfortunately, we found that these were not similar; for example, on day 10, the molar ratios of Pyr:2-OG:Glu were 1: 5.1:572.0 (intracellular) and 1:1.6:13.0 (extracellular). However, the molar concentration was Pyr<2-OG<Glu regarding both intracellular and extracellular at all the time points. We assume that further experiments using mutants of the mechanosensitive channels are necessary for conclusion.

20. Line 289: this is true, but according to your model in Fig. 4, Glu can be only made when ammonia is produced from nitrate, i.e. under HN conditions in the present study!

Response:

We thank the reviewer for this comment. We added “when nitrogen source is available” to correct the sentence (Line 347–348).

21. Line 296: I think the authors mean hyperosmotic

Response:

We thank the reviewer for this comment. We mean here hypoosmotic, actually (Line 353–355). We hypothesize that the mutant was in hypotonic conditions because, in comparison to the wild-type, it accumulated carbons as low-molecular-weight metabolites as opposed to high-molecular-weight glycogen.

REVIEWERS' COMMENTS:

Reviewer #1 (Remarks to the Author):

The updated version of the manuscript from Kato and colleagues has been substantially improved following the suggestions of all three reviewers. The authors have satisfactorily addressed all my concerns in this new version of the manuscript. The new modifications clearly add the extra information needed to fully understand the article. I have some minor comments and/or typos that I found during the revision of the document. All these comments are listed below.

- 1) The symbol " Δ " used to indicate a knockout should not be in italics. Please, correct it all the times it appears in the document.
- 2) Lines 111-112 and 277 – There is a missing word after "remained". Were you trying to say that nitrate remained available?
- 3) Line 194 – It is the first time that "AcCoA" that this molecule appears in the text. Please, write its full name and indicate the acronym between parentheses.
- 4) Line 242 – It is the first time that "Gln" that this molecule appears in the text. Please, write its full name and indicate the acronym between parentheses.
- 5) Line 303 – What is the function of the gene *sll0783*? Please, explain it here.
- 6) Lines 325-328 – Rephrase this sentence. It is complicated to read it.
- 7) Lines 330 – 332 - The name of the genes (*Synpcc7942_XXXX*) should not be in italics.
- 8) Lines 332 – 333 – What is the name of the mechanosensitive channels YbiO, MscM, MscS and MSck in *E. coli*? Please, indicate it here as it is included in the report for the reviewers.

Reviewer #2 (Remarks to the Author):

The authors have satisfactorily addressed my previously raised points. They have carefully revised the manuscripts and improved it with clearer representation of the results and additional data (Fig. 2 and Fig. 1B) and with extended discussion. I have no additional comments

Reviewer #3 (Remarks to the Author):

none

KOBE UNIVERSITY

Tomohisa Hasunuma
Engineering Biology Research Center
1-1 Rokkodai, Nada, Kobe 657-8501, Japan
Tel. +81-78-803-6202 Fax. +81-78-803-6102
E-mail: hasunuma@port.kobe-u.ac.jp

Responses to comments

Reviewer #1 (Remarks to the Author):

The updated version of the manuscript from Kato and colleagues has been substantially improved following the suggestions of all three reviewers. The authors have satisfactorily addressed all my concerns in this new version of the manuscript. The new modifications clearly add the extra information needed to fully understand the article. I have some minor comments and/or typos that I found during the revision of the document. All these comments are listed below.

1) The symbol “ Δ ” used to indicate a knockout should not be in italics. Please, correct it all the times it appears in the document.

Response:

We corrected the symbol “ Δ ” in the manuscript not to be in italics.

2) Lines 111-112 and 277 – There is a missing word after “remained”. Were you trying to say that nitrate remained available?

Response:

We revised the sentences as “nitrate in the medium remained available”.

3) Line 194 – It is the first time that “AcCoA” that this molecule appears in the text. Please, write its full name and indicate the acronym between parentheses.

Response:

We added the full name and indicated the acronym in the sentence.

4) Line 242 – It is the first time that “Gln” that this molecule appears in the text. Please, write its full name and indicate the acronym between parentheses.

Response:

We added the full name and indicated the acronym in the sentence.

5) Line 303 – What is the function of the gene sll0783? Please, explain it here.

KOBE UNIVERSITY

Tomohisa Hasunuma
Engineering Biology Research Center
1-1 Rokkodai, Nada, Kobe 657-8501, Japan
Tel. +81-78-803-6202 Fax. +81-78-803-6102
E-mail: hasunuma@port.kobe-u.ac.jp

Response:

We added the information of the *sll0783* gene in the sentence.

6) Lines 325-328 – Rephrase this sentence. It is complicated to read it.

Response:

We rephrased the sentence.

*7) Lines 330 – 332 - The name of the genes (*Synpcc7942_XXXX*) should not be in italics.*

Response:

We revised the manuscript so that the names of genes (*Synpcc7942_XXXX*) are not written in italics.

8) Lines 332 – 333 – What is the name of the mechanosensitive channels YbiO, MscM, MscS and MscK in E. coli? Please, indicate it here as it is included in the report for the reviewers.

Response:

The name of YbiO, MscM, MscS, and MscK in *E. coli* are as follows;

YbiO: moderate conductance mechanosensitive channel (NP_415329)

MscM: miniconductance mechanosensitive channel (NP_418583)

MscS: small conductance mechanosensitive channel (NP_417399)

MscK: potassium dependent, small conductance mechanosensitive channel (NP_414998)